# Bessel Equivariant Networks for Inversion of Transmission Effects in Multi-Mode Optical Fibres

**Joshua Mitton**[*]
School of Computing Science
j.mitton.1@research.gla.ac.uk

**Simon Peter Mekhail**
School of Physics & Astronomy,
Simon.Mekhail@glasgow.ac.uk

**Miles Padgett**
School of Physics & Astronomy,
Miles.Padgett@glasgow.ac.uk

**Daniele Faccio**
School of Physics & Astronomy,
Daniele.Faccio@glasgow.ac.uk

**Marco Aversa**
School of Computing Science
marco.aversa@glasgow.ac.uk

**Roderick Murray-Smith**
School of Computing Science
roderick.murray-smith@glasgow.ac.uk

University of Glasgow, Glasgow, Scotland, UK.

## Abstract

We develop a new type of model for solving the task of inverting the transmission effects of multi-mode optical fibres through the construction of an $\mathrm{SO}^+(2,1)$-equivariant neural network. This model takes advantage of the of the azimuthal correlations known to exist in fibre speckle patterns and naturally accounts for the difference in spatial arrangement between input and speckle patterns. In addition, we use a second post-processing network to remove circular artifacts, fill gaps, and sharpen the images, which is required due to the nature of optical fibre transmission. This two stage approach allows for the inspection of the predicted images produced by the more robust physically motivated equivariant model, which could be useful in a safety-critical application, or by the output of both models, which produces high quality images. Further, this model can scale to previously unachievable resolutions of imaging with multi-mode optical fibres and is demonstrated on $256 \times 256$ pixel images. This is a result of improving the trainable parameter requirement from $\mathcal{O}(N^4)$ to $\mathcal{O}(m)$, where $N$ is pixel size and $m$ is number of fibre modes. Finally, this model generalises to new images, outside of the set of training data classes, better than previous models.

## 1 Introduction

Multi-mode fibres (MMF) have many potential applications in medical imaging, cryptography, and communications. In the medical domain, the use of multi-mode fibre imaging has potential to create hair-thin endoscopes for imaging sensitive areas of the body. However, to achieve these applications, the fibre transmission properties must be compensated for to return a clear image (Stasio, 2017). A MMF has multiple different fibre modes, each of which propagates at a different velocity. This leads to an amplitude and phase mixing of the image as it propagates through the fibre (Mitschke, 2016). As a result, an input image creates a complex-valued speckled pattern on the output of the MMF. The ability to accurately and in a scalable way learn to invert the transmission effects would unlock MMF imaging as a useful tool across a range of domains. This work concerns the use of a single multi-mode fibre and not fibre bundles.

---

[*]https://github.com/JoshuaMitton

36th Conference on Neural Information Processing Systems (NeurIPS 2022).

Inverting a speckled image is challenging for multiple reasons. Firstly, the speckled images have a non-local relationship with respect to the original images. As a result, solely local patch-based models, such as convolutional neural networks, do not make sense as a solution without some dense mapping function. Therefore, the non-locality necessitates mapping the speckled images into a spatial arrangement similar to the original images before typical image-based deep learning techniques can be used, such as convolutions and pooling. In addition, the speckled images have circular correlation, which could be taken advantage of, although as noted by Moran et al. (2018), finding these requires solving the inversion, creating a chicken-and-egg problem. Finally, the fibre is equivalent to an unknown complex transmission matrix (TM) so, the inverse of this could be found using a complex-valued linear model, although this presents challenges in terms of memory requirements. A mapping between $350 \times 350$ original and speckled images would result in a TM with $350^4 \approx 15$ billion entries, requiring a linear model with as many parameters.

Previous work in inverting the transmission effects of MMFs has either required extensive experimentation to characterise the TM of the fibre (Čižmár & Dholakia, 2011, 2012; Choi et al., 2012; Mahalati et al., 2013; Papadopoulos et al., 2012; Plöschner et al., 2015; Leite et al., 2021), where the number of experimental measurements required for re-calibration was reduced by Li et al. (2021) by exploiting sparstiy in the TM; made use of dense linear models (Moran et al., 2018; Fan et al., 2019; Caramazza et al., 2019); or made use of convolutional models (Borhani et al., 2018; Rahmani et al., 2018). For the machine learning approaches to tackling the inversion task, those which make use of a dense linear model (Moran et al., 2018; Fan et al., 2019; Caramazza et al., 2019) can naturally account for the difference in spatial arrangement between speckled and original images, although they scale badly with the resolution of the images considered ($\mathcal{O}(N^4)$ for $N \times N$ resolution). On the other hand, in theory, the convolutional neural network (CNN) models (Borhani et al., 2018; Rahmani et al., 2018) improve upon the scalability issue, but in practice due to the need to approximate the transmission matrix and its non-local effects, the models require a large number of layers in order to be able to effectively map every pixel in the speckled image to every pixel in the original image, and hence in practice do not over come the scalability issues. All of these approaches are mostly expected to work for classes of objects that belong to the class that was used for training (Borhani et al., 2018), with Rahmani et al. (2018) making initial steps towards general imaging and Caramazza et al. (2019) demonstrating this for a more diverse testing dataset. We provide further details on each of the previous methods in Appendix A.2.

In this work we present a model which naturally accounts for the difference in spatial arrangement between speckled and original images and scales more efficiently than previous methods to higher resolution images. We believe this is the first method to demonstrate an ability to invert $256 \times 256$ pixel speckled images into $256 \times 256$ pixel original images. Our approach also takes advantage of the circular correlations in the speckled images, and improves upon previous general imaging results. Concerning the equivariance literature, we develop a model comprising of cylindrical harmonic basis functions, a basis set which has seen little attention in the equivariance literature, and make the connection between the transmission of light through a fibre and the group theoretic understanding used in developing equivariant neural networks. Our contributions are:

1. A more data-efficient, scalable model to solve the inversion of MMF transmission effects.
2. A model that provides better generalisation to out-of-training domain images.
3. A connection between group theoretic equivariant neural networks and the inversion of MMF transmission effects, providing a new type of model to tackle the problem.

## 2 Background

### 2.1 Multi-Mode Fibres

Multi-mode fibres present a clear advantage over single-mode fibre bundles due to having 1-2 orders of magnitude greater density of modes than a fibre bundle (Choi et al., 2012). Despite this, the different propagation velocities of each mode, which result in the fibre producing scrambled images, presents a significant challenge. If each propagation mode and velocity was known the TM could be computed, providing a linear system that inverts the transmission effects, although in general this is not known. Further, mode specific losses and imperfect mapping between the input pixels and fibre modes can lead to information loss such that inverting the transmission does

not yield the true original input image, as is demonstrated in Figure 1. Therefore, a model which correctly models the physics of inverting the transmission effects should produce the inverted image in Figure 1 and producing the original image requires addition information to be learned. We discuss the propagation of light through optical fibres and how we construct theoretical TMs in Section 2.2 and further details on the inversion of the TM in Appendix A.4.

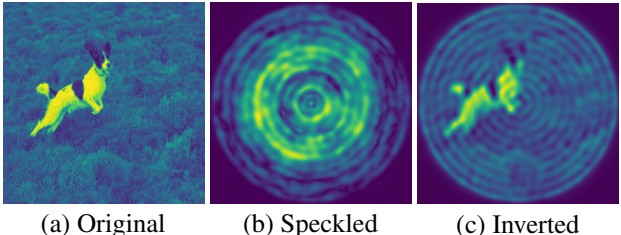

(a) Original     (b) Speckled     (c) Inverted

Figure 1: (a) Original image. (b) Speckled image from passing original images through a theoretical TM. (c) Inverted image created by passing speckled images through the inverted theoretical TM.

## 2.2 Generation of Theoretical Transmission Matrices

Light propagation through a MMF is characterised by the transmission modes of the fibre. The fibres considered in this work have a step index profile with the refractive index within the core being constant and at a higher value than the cladding. Analytical solutions for the fibre modes can be determined by solving the Helmholtz equation in cylindrical coordinates (details in Appendix A.1). The problem can therefore be expressed as an eigenvalue eigenfunction problem, where the eigenfunctions are the fibre modes and the eigenvalues are the propagation constants, $\beta$, of the modes. The solutions for the electric field within the fibre are comprised of Bessel functions of the first kind inside the core and modified Bessel functions of the second kind in the cladding as follows:

$$f_l^{core}(r,\theta) = \frac{J_l(u(\beta) \cdot \frac{r}{R})}{J_l(u(\beta))}e^{\pm il\theta}, \qquad f_l^{clad}(r,\theta) = \frac{K_l(w(\beta) \cdot \frac{r}{R})}{K_l(w(\beta))}e^{\pm il\theta}, \qquad (1)$$

where $r$ and $\theta$ are the radial and azimuthal coordinates, respectively, $l$ is the azimuthal index of the mode, $R$ is the fibre core radius, and $u$ and $w$ are normalised frequencies defined as follows:

$$u(\beta) = R\sqrt{k_0^2 n_{core}^2 - \beta_{lm}^2}, \qquad w(\beta) = R\sqrt{\beta_{lm}^2 - k_0^2 n_{clad}^2}, \qquad (2)$$

where $n_{core}$ and $n_{clad}$ are the refractive indices of the core and cladding, respectively, $k_0$ is the vacuum wave number, and $m$ is the radial mode index. Further details on the connection between propagation of light through MMFs and group theoretic equivariant neural networks are provided in Appendix A.1. Taking the derivative of Equations 1, and equating the resulting functions asserts a smoothness condition on the electric field across the core-cladding boundary. Solving for this for all values of $\beta$ over all possible integer values of $l$ gives the propagation constants. These constants correspond to the fibre modes given by the now appropriately parameterised Equations 1. Examples of the mode fields are given in Figure 2.

A diagonal fibre propagation matrix, $F$, can then be made from the eigenvalues, $\beta$, of the Helmholtz problem and the corresponding eigenfunctions can be recorded in a matrix, $M$, which maps the image space to the fibre mode, or group space. We can find the inverse mapping bases from the fibre or group space back to the image space, $M^\dagger$, by taking the conjugate transpose. These three components allow us to construct the transmission matrix as: $\text{TM} = MFM^\dagger$.

## 2.3 Equivariance

Equivariance in neural networks concerns the study of symmetries and uses these to build an inductive bias into a model (Cohen & Welling, 2016a; Bronstein et al., 2021). Equivariance as a property of a neural network guarantees that a transformation of the input data produces a predictable transformation of the predicted features (Worrall et al., 2017). Formally, we say that, given two transformations $T$ and $T'$ which are linear representations of a group $G$ and a network or layer $\Phi$, the network is equivariant if applying transform $T_g$ to an input $x$ and then passing it through the

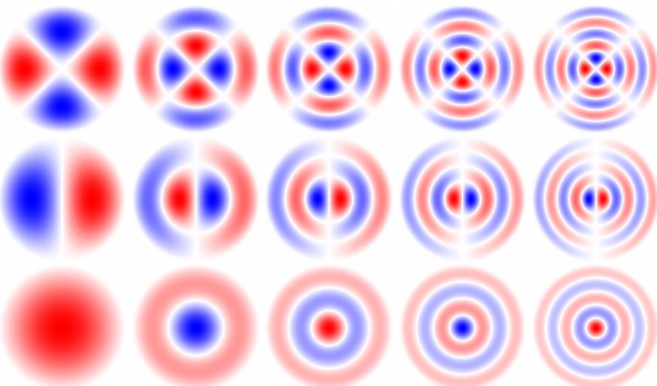

Figure 2: Electric field amplitude profiles of the Bessel bases used within our model. Here red is positive, blue is negative and the colour saturation represents the electric field strength. These bases are often known as LP modes.

network $\Phi$ yields the same result as first passing $x$ through $\Phi$ and then transforming by $T'_g$ (Cohen & Welling, 2016a); that is,

$$\Phi T_g(x) = T'_g(\Phi(x)). \tag{3}$$

Convolutional neural networks (CNNs) are an early example of equivariance on images, where by design they are translationally equivariant (LeCun et al., 1998). More recently, equivariance has been considered for other group actions than that of the translation group. The majority of works on images involve incorporating rotation or reflection equivariance into the model, where a variety of different group transformations have been considered (Cohen & Welling, 2016a,b; Weiler & Cesa, 2019; Murugan et al., 2019; Wiersma et al., 2020; De Haan et al., 2020; Franzen & Wand, 2021).

We are not interested in developing a group equivariant convolutional model here due to the non-similar spatial arrangements between the speckled and original image domains. Despite this, the concept of equivariance still applies, as the concept of learning a model from a fixed basis set, which guarantees symmetry properties are conserved, is relevant due to the nature of transmission through multi-mode optical fibres.

## 2.4 Circular Harmonics

Of the works that incorporate rotation equivariance into CNNs we are particularly interested in those using a continuous rotation group $SO(2)$. In the case of discrete rotation groups a different feature space is associated to each rotation angle. Storing features for an infinite number of rotation angles is not computationally tractable. One option is to turn to a Fourier representation, whereby instead of choosing a discrete number of rotations a maximum frequency can be chosen. To use a Fourier representation for the weights of the model the domain $\mathbb{R}^2$ is decomposed into a radial profile and angular function. Solving for the kernel space of permissible filters that can be used within a CNN and ensuring that the model maintains rotation equivariance yields only the spectrally localised circular harmonics (Worrall et al., 2017; Weiler & Cesa, 2019; Franzen & Wand, 2021). Therefore a rotation equivariant CNN can be created by solving for the circular harmonics up to a certain rotational frequency, combining this with a radial profile function, and sampling a basis of resolution given by the CNN filter size. This yields a set of bases which can be linearly combined with a learnable weighting applied to each to form the kernel for the CNN.

### 2.4.1 Cylindrical Harmonics

While the circular harmonics have seen some attention in the deep learning community due to the use in constructing rotational equivariant CNNs, the cylindrical harmonics have seen less attention (Klicpera et al., 2020). The cylindrical harmonics appear as a solution to Bessel functions for integer $\alpha$ and are therefore of interest for problems where information transformation is characterised by such functions. For example Bessel functions are used when solving wave or heat propagation.

Bessel functions are solutions for different complex numbers $\alpha$ of Bessel's differential equation:

$$x^2 \frac{d^2 y}{dx^2} + x \frac{dy}{dx} + (x^2 - \alpha^2) y = 0. \tag{4}$$

For integer values of $\alpha$ the Bessel function solutions are a linearly independent set of functions expressed in cylindrical coordinates. Each function consists of the product of three functions. The radially dependent term is typically called the cylindrical harmonics. Further details on the connection between group theory and the Bessel basis functions is provided in Appendix A.1.

## 3 Our Method

Our method comprises two models: a Bessel equivariant model and a convolutional post-processing model. The Bessel equivariant model uses a basis set comprising cylindrical harmonics and learns a complex mapping function to use the known symmetries in optical fibres. The Bessel equivariant model is constrained by the basis choice to produce circular images, which can have circular artifacts also seen in a true inversion, see Figure 1. The post-processing model is used to fill gaps in the corners, remove circular artifacts, and sharpen the images. The two stage approach is useful in safety-critical applications as users can inspect the output of both models, where the Bessel equivariant model output is less likely to depend on the training data due to the choice of inductive bias.

### 3.1 Bessel Basis Equivariant Model

The core of our model has an inductive bias that better replicates the physics of the task of inverting the transmission effects of MMF imaging. To achieve this we utilise prior knowledge of MMFs and the modes with which an image propagates through the fibre. Similar to the successes of rotation equivariant neural networks, which make use of circular harmonics to achieve rotation equivariance, we utilise cylindrical harmonics as an inductive bias of the model. This is motivated by the propagation modes of the fibre being characterised by Bessel functions, where the radially dependent solution is given by the cylindrical harmonics. We make the connection to the group theoretic development of equivariant neural networks in Appendix A.1, demonstrating the connection between the group $\mathrm{SO}^+(2, 1)$, the light cone, and Bessel function solution to the wave equation.

The first stage of the model is the computation of two sets of basis functions. The first set transforms the input image into a function that lives on the group and the second transforms back into the image domain. The model's weight matrix linearly transforms the function mapped by the first basis set. The basis filters are computed in accordance with equations in Appendix 2.2.

These functions are sampled on a grid given by the size of the speckled images for the first basis set and the original images for the second basis set. An example of the bases produced is given in Figure 2. The Bessel function bases are computed in a pre-training stage and therefore this step only has to be completed once when creating the model. Further, these bases are not trainable parameters and are not updated during training of the model. These basis functions are an alternative to those used in general rotation equivariant neural networks, which typically offer equivariance to the group $\mathrm{SO}(2)$, except they correctly model the symmetry group, $\mathrm{SO}^+(2, 1)$ of the task of inverting transmission effects of a MMF due to the added time dimension.

The model trainable parameters are complex-valued weights of size equal to the number of bases. The complex weight matrix is a diagonal matrix. The images transformed by the first basis set are multiplied by the weight matrix before being transformed by the second basis set to produce the predicted image. The overall model architecture is given in Figure 3.

### 3.2 Post-Processing Model

As demonstrated in Section A.4 the limited number of modes means that the transmission effects of a MMF are typically not fully invertible leading to information loss. We would therefore not expect an equivariant model informed by the physics of the task to be able to fully invert the transmission effects. To overcome this we add a second model which is similar to a super-resolution model, except that it does not change the resolution and instead learns to remove circular artifacts, fill in gaps, and sharpen the images. It is a fully convolutional model comprising of convolutional layers and non-linearities. This model takes as input the output of our Bessel equivariant model, which we suspect

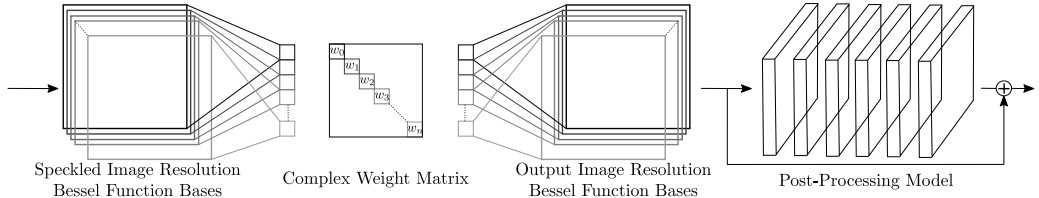

Speckled Image Resolution Bessel Function Bases     Complex Weight Matrix     Output Image Resolution Bessel Function Bases     Post-Processing Model

Figure 3: Our two stage model architecture comprising of a Bessel equivariant model and post-processing model. Input (left) is speckled images. Output (right) is predicted image. The speckled image resolution Bessel function bases transform the speckled image into a mode space of the group $SO^+(2, 1)$. The complex weight matrix is diagonal. The original image resolution Bessel function basis provides a mapping from the mode space to the output image space. The trainable parameters of the Bessel equivariant model are the diagonal complex weight matrix only. The post-processing model is a convolutional model.

will be similar to what is achievable by passing the speckled images through an inverted TM, and predicts the original images. This model is composed of eight convolutional layers with ReLU non-linearities. The motivation behind the explicit separation of the inversion and enhancement models is that in some applications, especially safety-critical applications, it is important to be able to see the results of an inversion conditioned on the observations alone, regardless of prior expectations.

## 4 Experiments

We investigate performance with both data generated with a theoretical TM and experimental data collected using a real MMF fibre. We provide details of the theoretical TM generation in Section 2.2. We use the data from a real fibre, as described in Moran et al. (2018), to validate that our new approach can model the same data, despite using significantly fewer trainable model parameters. Further, we generate data using a theoretical TM as this allows us to flexibly adapt the parameters of the experiment and create new datasets to validate the new model design. Finally, the theoretical TM allows us to experiment with images with higher resolution than has been previously been considered. We provide details on the training of the models and datasets in Appendix A.3.

### 4.1 Real Multimode Fibre

We compare our model to the complex-valued linear model developed by Moran et al. (2018) by considering both MNIST and fMNIST data. For this we train and test separate models for the MNIST and fMNIST datasets. In addition, we compare multiple different versions of our model where the diagonal restriction of the mapping between bases is relaxed to allow for a block diagonal structure. We do this to account for manufacturing defects, sharp bends, dopant diffusion, elliptical cores which causes the diagonal mapping of the fibre matrix within the TM to be block diagonal (Carpenter et al., 2014). This relaxation allows for $x$-offsets above and below the main diagonal.

Table 1 shows that our model with a full mapping matrix between bases, i.e. full relaxation of the diagonal constraint, outperforms all other models. In addition, with a relaxation to allow for a 10 block diagonal structure our model performs comparably. On the other hand, the complex linear model outperforms our Bessel equivariant model when the diagonal restriction on the mapping function between bases is enforced. Despite this, our model still provides clear and accurate reconstructions of

Table 1: Comparison of the loss values of each model trained with MNIST or fMNIST data.

| Model | MNIST | | fMNIST | |
| --- | --- | --- | --- | --- |
| | Train Loss | Test Loss | Train Loss | Test Loss |
| Complex Linear | 0.00396 | 0.00684 | 0.00509 | 0.01061 |
| Bessel Equivariant Diag | 0.03004 | 0.03125 | 0.02903 | 0.03140 |
| Bessel Equivariant Diag + Post Proc | 0.01317 | 0.01453 | 0.01609 | 0.01749 |
| Bessel Equivariant 10 Off Diag + Post Proc | 0.00488 | 0.00576 | 0.00943 | 0.01162 |
| Bessel Equivariant Full | 0.00300 | 0.00684 | 0.00548 | 0.01380 |
| Bessel Equivariant Full + Post Proc | **0.00145** | **0.00378** | **0.00306** | **0.00964** |

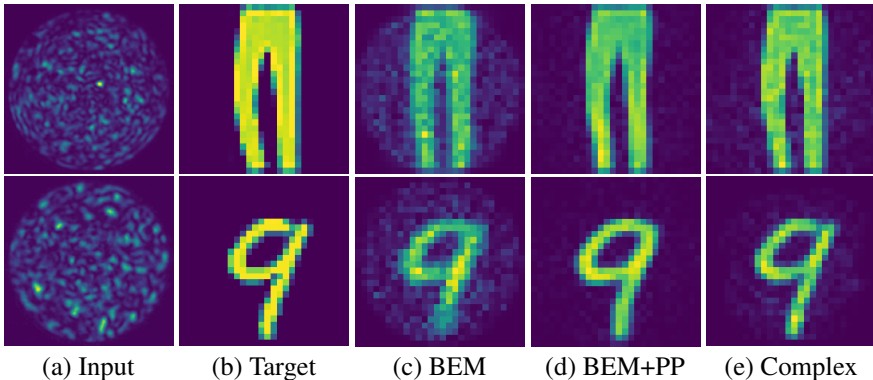

|      (a) Input      |      (b) Target      |      (c) BEM      |      (d) BEM+PP      |      (e) Complex      |

Figure 4: Comparison of predicted images from inverting transmission effects of a MMF. The upper row is fMNIST data, lower row MNIST data. (a) Input speckled image, (b) the target original image to reconstruct, (c) Output of the Bessel equivariant model, (d) Output of the combination of Bessel equivariant and post-processing model, and (e) the output of the complex-valued linear model.

the target images whilst using orders of magnitude fewer trainable parameters. Additional results are presented in Tables 6 and 7. Further to a comparison of the loss values, we visually inspect the reconstruction quality in Figure 4, and we provide further visualisations in Figures 16 and 17. Thus despite our Bessel equivariant diagonal model achieving a larger loss value, the digit or item is still clearly visible in the prediction and is correctly sharpened to a realistic prediction of the original image by our post-processing model. The complex linear model can also be seen to predict a realistic looking output of the correct digit and object, although some noise does feature in the prediction. In addition, we visually compare each of our Bessel equivariant models with the diagonal restriction relaxed in Figures 10 and 11, which demonstrates that as the diagonal restriction is relaxed the model can better predict the original images with less noise in the prediction from the Bessel equivariant model.

We also compare the number of trainable parameters within the model in Table 2. This highlights that our model requires two orders of magnitude fewer parameters that the complex linear model when using the diagonal Bessel equivariant model. Therefore, our model has the potential to scale to higher resolution images where the complex linear model will run out of GPU memory. As the diagonal mapping between Bessel bases is relaxed to include 10-block diagonal structure the model still requires two order of magnitude fewer parameters. In the most flexible version of our model the number of trainable parameters is comparable with the complex Linear model, although we have demonstrated that this level of flexibility is not required to achieve good image reconstruction.

Table 2: Comparison of # trainable parameters in each model trained with MNIST or fMNIST data.

| Model | Number of Trainable Parameters (Millions) |
|---|---|
| Complex Linear | 78.826 |
| Bessel Equivariant Diag + Post Proc | 0.500 |
| Bessel Equivariant 10 Off Diag + Post Proc | 0.617 |
| Bessel Equivariant Full + Post Proc | 43.108 |

## 4.2 Theoretical TM

We compare our model to the complex linear model developed by Moran et al. (2018) with data created using a theoretical TM. We create multiple datasets one using $28 \times 28$ fMNIST images, with $180 \times 180$ speckled images, another with MNIST images, of the same size, and one with a subset of images from the ImageNet dataset (Deng et al., 2009) where we fix the resolution to be $256 \times 256$ of both the images and speckled images. Using fMNIST and MNIST with a theory TM is useful for developing understanding, due to the ease of creating different experiments and datasets, and for testing the generalisability of the model between the two datasets. We include the ImageNet-based dataset as these are higher resolution images containing more challenging information than fMNIST. To the best of our knowledge, demonstrating the ability to invert transmission effects of such high resolution images has not been previously achievable with a machine learning based approach.

Table 3: Comparison of the loss values of each model trained with fMNIST data.

| Model | fMNIST Train Loss | fMNIST Test Loss | MNIST Test Loss |
|---|---|---|---|
| Complex Linear | 0.0149 | 0.0146 | 0.0363 |
| Bessel Equivariant | 0.0141 | 0.0139 | **0.0026** |
| Bessel Equivariant + Post Proc | **0.0032** | **0.0032** | 0.0028 |

#### 4.2.1 Generalisability of Models

We provide a comparison of the predictions of a complex valued linear model (Moran et al., 2018), our equivariant model only, and our equivariant model coupled with a post-processing model trained on fMNIST images and tested on both fMNIST and MNIST to analyse to what extent the models can generalise to a new dataset. Table 3 presents the loss values for both the training and testing of each model. Considering the fMNIST data this shows that the complex linear and Bessel equivariant models achieve similar loss values. Although this is a result of the Bessel equivariant model under-performing due to not being able to reconstruct pixel values close to the edge of the image as a consequence of the circular nature of the Bessel bases functions, while the complex linear model under-performs due to failing to reconstruct higher frequency information and over-fitting to the general clothing categories. Finally, when our Bessel equivariant model is combined with the post-processing model it outperforms all other models. The ability to generalise to a new data is assessed through the MNIST dataset. This shows that our Bessel equivariant model significantly out-performs the complex linear model in generalising to a new dataset.

Figure 5 shows reconstructions produced by the three models, with further reconstructions provided in Figure 18. This demonstrates visually that our model, even without the post-processing part, is able to better reconstruct the original images. The post-processing model refines the output of the equivariant model and fills in gaps outside the fibre. Despite it being possible to estimate the general category of clothing from the outputs of the complex linear model, the higher frequency information has been lost, i.e. the pattern on the jumper or the curve in the trouser leg. On the other hand, our model is able to reconstruct this higher frequency information.

Previous works have demonstrated some ability to generalise to new data classes (Rahmani et al., 2018; Caramazza et al., 2019), although in each case the results are not perfect. Here, we compare the different methods trained on fMNIST and then tested on MNIST data. Figure 6 demonstrates that our Bessel equivariant model generalises well to a new data domain, with further reconstructions provided in Figure 19. On the other hand, the complex linear model somewhat predicts the correct digits. In this out of training domain our post-processing model does not add any value to the original Bessel equivariant model. This highlights the benefit of our two-stage modelling approach as we have one robust model and a second model to sharpen the images, as a result if a user believes the situation could be unusual they could reliably use the output of our Bessel equivariant model and not the post-processing part. Further to achieving strong generalisation results, we also consider the effect of noise in the speckled images in Appendix A.6.

Figure 8 shows how the number of trainable model parameters scale. Even for $28 \times 28$ images our model requires two orders of magnitude fewer trainable parameters than the complex linear model.

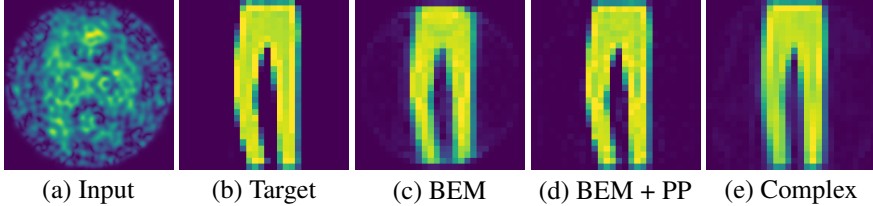

| (a) Input | (b) Target | (c) BEM | (d) BEM + PP | (e) Complex |

Figure 5: Comparison of predicted images from inverting transmission effects of a MMF. (a) Input speckled image, (b) Target original image to reconstruct, (c) Output of the Bessel equivariant model, (d) Output of the Bessel equivariant and post-processing model, and (e) Output of the complex valued linear model.

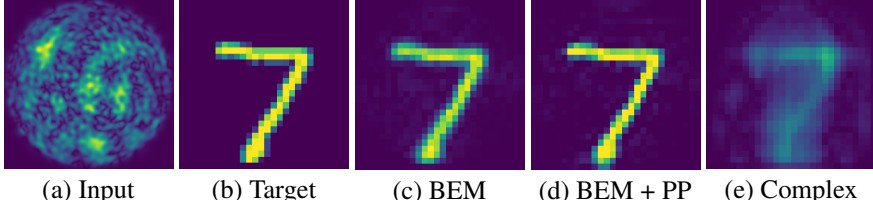

| (a) Input | (b) Target | (c) BEM | (d) BEM + PP | (e) Complex |

Figure 6: Comparison of predicted images from inverting transmission effects of a MMF. (a) Input speckled images, (b) Target original images, (c) Output of the Bessel equivariant model, (d) Output of Bessel equivariant and post-processing model, and (e) Output of the complex valued linear model.

### 4.2.2 Scaling to Larger Images – ImageNet

We now experiment with a subset of images from the ImageNet dataset, where we fix the resolution to be $256 \times 256$. Inverting the transmission effects of such higher resolution images has not been previously achievable. We only compare our equivariant model with our equivariant model coupled with a fully convolutional post-processing model. The complex-valued linear model cannot fit in the GPU memory of an A6000 with 48.7GiB of memory with this resolution of image.

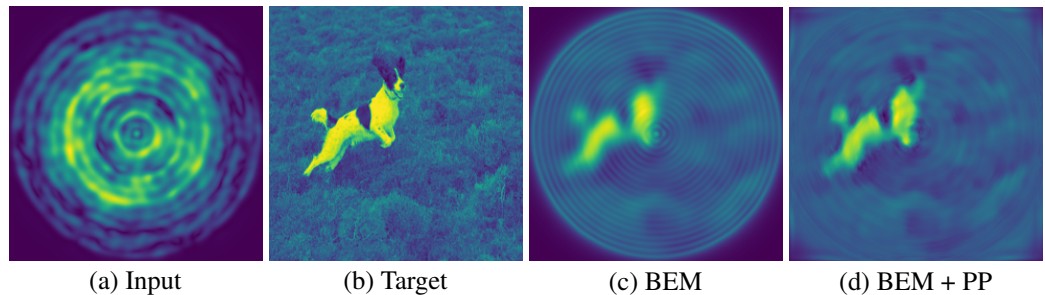

| (a) Input | (b) Target | (c) BEM | (d) BEM + PP |

Figure 7: Comparison of predicted images from inverting transmission effects of a MMF using high resolution ImageNet data. (a) Input speckled image, (b) Target original image to reconstruct, (c) Output of Bessel equivariant model (d) Output of Bessel equivariant and post-processing model.

We analyse our model visually, with Figure 7 showing that our Bessel equivariant model produces a reconstruction from which one can determine the type of dog and its activity, although the reconstruction does not capture all the high frequency information, and information in the corners is lost due to the circular nature of the fibre. Further reconstructions are provided in Figure 20. When we combine the two models some of the artifacts of the Bessel equivariant model are removed as the post-processing model fills in information towards the corners and sharpens the image.

We present the loss values for both models in Table 4. This shows a similar result as for fMNIST, that our Bessel equivariant model can solve the task well, but requires the post-processing model to sharpen the image and fill in gaps in the corners. We compare the models' memory requirements in Figure 8. Note how fewer trainable parameters translate into significantly less memory use than the complex linear model. This effect is seen more drastically when scaling to high resolution images, where the complex linear model runs out of memory on a 24.2GiB Titan RTX for original and speckled images of resolution $180 \times 180$ pixels. Our model, in contrast, has been tested on $256 \times 256$ pixel images, requiring only 2.1GiB, and can scale to megapixel images.

Finally, we explore the impact on each model of reducing the size of the training dataset. Requiring a reduced dataset size minimises the time taken in a lab collecting data. Figure 8 demonstrates that our model outperforms the complex linear model with $10\times$-less training data, reducing the need to collect 12000 samples to 1200. The reconstructions are visualised in Appendix A.12.

Table 4: Comparison of the loss values of each model trained with ImageNet data.

| Model | Train Loss | Test Loss |
|---|---|---|
| Bessel Equivariant | 0.0541 | 0.0574 |
| Bessel Equivariant + Post Proc | **0.0161** | **0.0159** |

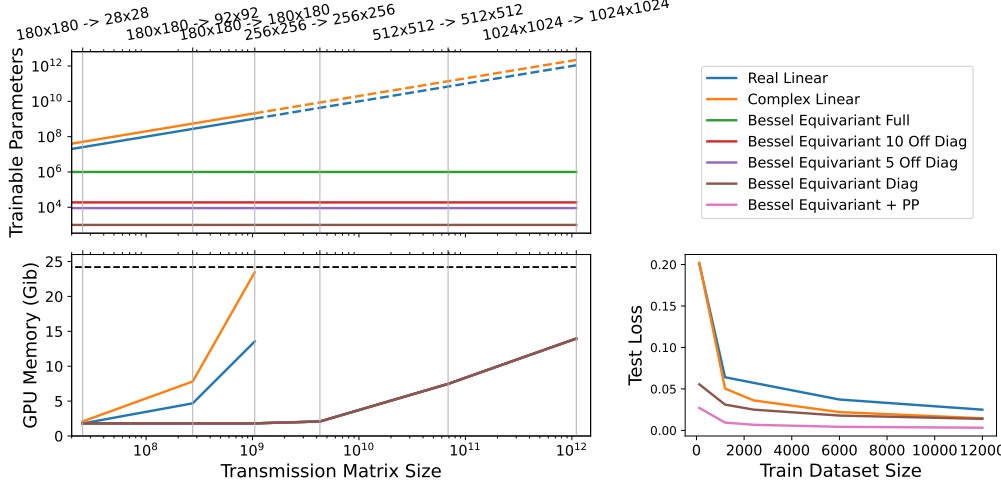

Figure 8: (Upper Left) Comparison of number of trainable parameters v. TM size (dashed line indicates model cannot in practice be built due to large memory requirements). (Lower Left) Comparison of required GPU memory against TM size. Vertical grey lines indicate a given image size – our model can process $1024 \times 1024$ images on a single consumer-level Titan RTX GPU. (Lower Right) Comparison of reducing the size of the training dataset. All plots for a 1000 mode fibre.

## 5    Conclusions

We present a new type of model for solving the task of inverting the transmission effects of a MMF through developing a $\mathrm{SO}^+(2,1)$-equivariant neural network and combining this with a post-processing network. This improves upon complex linear models through incorporating a useful physically informed inductive bias into the equivariant network. The equivariant network is shown to perform well on new image classes, providing a general model for fibre inversion. The post-processing network is specific to an image domain, as it generalises to new image classes as well as a general convolutional network, although it improves the quality of returned images. The use of a theoretical TM allows us to compare an 'ideal' inverse with the output of previous models based on learning a full transmission matrix. This suggests that previous learned transmission matrices were combining elements of the actual transmission matrix with elements of the post-processing network, which would affect their ability to generalise to new images. We also anticipate that in interactive safety-critical applications users might want the ability to switch between modes, to be sure that the evidence was there, and not overly influenced by the priors in the training data.

This new model significantly improves the ability to scale to higher resolution images by improving the scaling law from $\mathcal{O}(N^4)$ to $\mathcal{O}(m)$, where $N$ is pixel size and $m$ is the number of fibre modes. We provide a comparison between our model and prior works on both data created using a real fibre and a theoretical TM, demonstrating in both cases that our model solves the task while using significantly fewer trainable parameters than the complex and real linear models. In addition, we demonstrate results on high resolution $256 \times 256$ images, which has previously been unachievable due to the growth of parameters with previous models. Furthermore, we demonstrate the ability of our model to generalise to new data classes outside of the training data, outperforming prior works. The dramatic reduction in the number of parameters for each fibre configuration opens the way for future models which can learn mappings for high-resolution images, from a wider set of perturbed fibre poses and combine these using architectures such as VAEs.

## Acknowledgments and Disclosure of Funding

JM is supported by a University of Glasgow Lord Kelvin Adam Smith Studentship. RM-S, MP, DF and SPM are grateful for EPSRC support through grants EP/T00097X/1 and RM-S for EP/R018634/1. MA is funded by dotPhoton and a UofG scholarship. DF acknowledges funding through the Royal Academy of Engineering Chair in Emerging Technologies programme.

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
