# A Appendix

## A.1 Group Theoretic Understanding of Optical Fibre Transmission Modes

When a light beam propagates in free space or in a transparent homogeneous medium, its transverse intensity profile generally changes. Despite this, there exist certain distributions that do not change intensity profile as they traverse. These fixed profiles are the transmission modes of the space.

The development of group equivariant networks exploits a similar principle, where these networks are constructed under the more general principle of finding basis functions called irreducible representations of some group. Some examples of this principle include: for the circle $S^1$ or line $\mathbb{R}$ the irreducible representations are given by complex exponentials $\exp(in\theta)$, for the group $SO(2)$ the irreducible representations are given by the circular harmonics, for the group $SO(3)$ the irreducible representations are given by the Wigner D-functions, and for $S^2$ the irreducible representations are given by the spherical harmonics. Thus a function on the group can be composed as a linear combination of the corresponding irreducible representations. On the other hand, when learning a function it is possible to learn the weightings of each of the irreducible representations of the group to learn that function. Building a model such that its feature space comprises the irreducible representations of the group provides a method of constructing a model that is equivariant to the underlying group of the representations.

Here we seek to show a connection between the known properties of optical fibres and group equivariant networks. We start by providing some details on the propagation of light through a fibre. The propagation of light is governed by the time-independent (Helmholtz) equation, which in cylindrical coordinates is given by Equation 5.

$$\frac{\partial^2 E}{\partial r^2} + \frac{1}{r}\frac{\partial E}{\partial r} + \frac{1}{r^2}\frac{\partial^2 E}{\partial \phi^2} + q^2 E = 0. \tag{5}$$

The standard approach to solving the above equation is to use the separation-of-variables procedure, which assumes a solution of the form given in Equation 6.

$$E_z = AF_1(r)F_2(\phi). \tag{6}$$

Due to the circular symmetry of the fibre, each component must not change when the coordinate $\phi$ is increased by a multiple of $2\pi$. Therefore, we make the following assumption:

$$F_2(\phi) = e^{i\nu\phi}, \tag{7}$$

where $\nu \in \mathbb{Z}$. Substituting into Equation 5 yields a wave equation of the following form:

$$\frac{\partial^2 F_1}{\partial r^2} + \frac{1}{r}\frac{\partial F_1}{\partial r} + \left(q^2 - \frac{\nu^2}{r^2}\right)F_1 = 0, \tag{8}$$

which is a differential equation for Bessel functions. Solving this both inside the core of the fibre and in the cladding of the fibre provides two solutions. In the core of the fibre, as $r \to \infty$ the guided modes must remain finite. Thus for $r \leq a$ for core radius $a$ the solution is a Bessel function of the first kind:

$$E_z(r \leq a) = AJ_\nu(ur)e^{i\nu\phi}e^{i(\omega t - \beta z)} \tag{9}$$

and outside of the core the solution is a modified Bessel function of the second kind:

$$E_z(r \geq a) = CK_\nu(wr)e^{i\nu\phi}e^{i(\omega t - \beta z)}. \tag{10}$$

Solving these equations provides the transmission modes of the fibre, i.e. the non-changing distributions.

Next we consider the propagation of light through a fibre from a group theoretic perspective. The indefinite special orthogonal group $SO(2,1)$ is the group considering two spatial dimensions and one time dimension, and can be realised as:

$$SO(2,1) = \{X \in \text{Mat}_3(\mathbb{R}) | X^t \nu X = \nu, \det(X) = 1\} \tag{11}$$

where,

$$\nu = \begin{pmatrix} 1 & 0 & 0 \\ 0 & 1 & 0 \\ 0 & 0 & -1 \end{pmatrix}. \tag{12}$$

To identify the Lie algebra $\mathfrak{so}(2,1)$ we use the tangent space $T_1 SO(2,1)$ to $SO(2,1)$ at the identity 1. We then choose a curve $a : L \to SO(21)$ such that $a'(0) = A$ Then, the characterising equation of $A$ gives:

$$a(t)^T \nu a(t) = \nu. \tag{13}$$

Taking the derivative with respect to $t$ gives:

$$a'(t)^T \nu a(t) + a(t)^T \nu a'(t) = 0. \tag{14}$$

Then, evaluating the expression at $t = 0$ gives:

$$A^T \nu + \nu A = 0. \tag{15}$$

Now we check the linear conditions determined by the above characterisation to then write out the Lie algebra, with $\nu$ having a natural block decomposition. Therefore, for a general element $A \in \mathfrak{so}(2,1)$, given as:

$$A = \begin{pmatrix} W & x \\ y^T & z \end{pmatrix}, \tag{16}$$

where $W \in M(2, \mathbb{R})$, $x, y \in \mathbb{R}^2$, and $z \in \mathbb{R}$. In this block decomposition $\nu = \begin{pmatrix} \mathbb{I}_2 & 0 \\ 0 & -1 \end{pmatrix}$. Then, Equation 15 becomes:

$$\begin{pmatrix} W^T & y \\ x^T & z \end{pmatrix} \begin{pmatrix} \mathbb{I}_2 & 0 \\ 0 & -1 \end{pmatrix} + \begin{pmatrix} \mathbb{I}_2 & 0 \\ 0 & -1 \end{pmatrix} \begin{pmatrix} W & x \\ y^T & z \end{pmatrix} = \begin{pmatrix} 0 & 0 \\ 0 & 0 \end{pmatrix}. \tag{17}$$

Therefore, the following conditions are imposed:

$$\begin{aligned} W^T &= -W \\ y &= x \\ z &= 0. \end{aligned}$$

Hence, the Lie algebra $\mathfrak{so}(2,1)$ is given by:

$$\mathfrak{so}(2,1) = \left\{ \begin{pmatrix} W & x \\ x^T & 0 \end{pmatrix} : W^T = -W \right\} = \left\{ \begin{pmatrix} 0 & -w & x_1 \\ w & 0 & x_2 \\ x_1 & x_2 & 0 \end{pmatrix} \right\}. \tag{18}$$

As a result we can see that the condition on $W$ is such that $W \in \mathfrak{so}(2, \mathbb{R})$. Therefore, the condition on $W$ characterises the rotational symmetry of the group. Given the goal is make the connection between the group theoretic understanding of group equivariant neural networks and solution to the wave equation in cylindrical coordinates, this is promising as the solutions to Bessel functions have rotational symmetries.

Further, the group $SO(2,1)$ is that of two spatial dimensions and one time dimension. The connection between the three spatial dimensional case and the view as a light cone was made in the development of Minkowski spacetime. Here you imagine a cone where the time axis runs from the point of a cone through the centre of the plane drawn on the open end and the two spatial axes form a plane which intersects the cone and is parallel to the plane drawn on the open end. This is known as the future light cone and is an interpretation of how light spreads out after an event occurs. The group actions of the group $SO^+(2,1)$, which is the group $SO(2,1)$ with the requirements that $t > 0$, act on this space and can be viewed as rotations of the three dimensional Euclidean sphere. A connection can be drawn between this view and the fact that non compact generators of $SO(n,1)$ differ from corresponding matrix elements of the same generators of $SO(n+1)$, the group of rotations in $n+1$-dimensional space, by a factor of $\sqrt{(-1)}$ (Wong, 1974). Therefore, the connection can be made between the group actions of $SO^+(2,1)$ and the light cone view of light propagation.

A final connection can be made between the light cone and the wave equation, given the connection between the group action and the light cone. Returning to the wave equation we note that it describes waves travelling with frequency independent speed. The character of the solution is different in odd and even dimensional spaces. In an odd dimensional space a disturbance propagates on the light cone and vanishes elsewhere. On the other hand, in an even dimensional space a disturbance propagates inside the entire light cone. Therefore, in an even dimensional space a disturbed medium

never returns to rest. This phenomena is known as geometric dispersion. Here we are interested in even dimensional space as we have two spatial dimensions governing transmission through the fibre along with a third time dimension. We therefore expect the propagation of light through the fibre to be understood by propagation inside the entire light cone. Given the wave equation:

$$u_{tt} = c^2(u_{xx} + u_{yy}).$$ (19)

A solution is possible by considering the three-dimensional theory, if we regard $u$ as a function in three dimensions and that the third dimension is independent. So, if we require:

$$\begin{aligned} u_0(0, x, y) &= 0 \\ u_t(0, x, y) &= \phi(x, y), \end{aligned}$$

then the three-dimensional solution equation becomes:

$$u(t, x, y) = tM_{ct}[\phi] = \frac{t}{4\pi} \iint_S \phi(x + ct\alpha, y + ct\beta) d\omega,$$ (20)

where $\alpha$ and $\beta$ are the first two coordinates on the unit sphere and $d\omega$ is the area element on the sphere. The integral can be written as a double integral over disc $D$ with centre $x, y$ and radius $ct$:

$$u(t, x, y) = \frac{1}{2\pi ct} \iint_D \frac{\phi(x + \xi, y + \nu)}{\sqrt{(ct)^2 - \xi^2 - \nu^1}} d\xi d\nu.$$ (21)

It becomes clear that the solution does not only depend on the data on the light cone where:

$$(x - \xi)^2 + (y - \nu)^2 = c^2 t^2,$$ (22)

but on the entire data inside the light cone. Therefore, it can be seen how the solution to the wave equation yields the interpretation of the light cone in the precise way that was yielded by the analysis of the group $\text{SO}^+(2, 1)$.

This completes the connections between the group action of $\text{SO}^+(2, 1)$ and the solution to the Bessel functions that govern the transmission modes within optical fibres. Now we can compose a model in a similar way to how circular harmonics are used to construct rotation equivariant neural networks under the group $\text{SO}(2)$, but with utilising the basis functions found by solving the Bessel function in Equation 8 to compose a network which is equivariant to the group $\text{SO}^+(2, 1)$. This is useful as it positions the model with respect to other group equivariant neural networks and provides a model with suitable inductive bias for the task of learning a function approximation to transmission through optical fibres.

## A.2 Related Work

One method for inverting the transmission effects is to find the underlying TM of the multi-mode fibre. In general this is not known, although in principle the TM can be found by acquiring the output amplitude and phase relative to each mode (Čižmár & Dholakia, 2011, 2012). Choi et al. (2012) present an approach to construct the TM in a scanner-free method based on measurement of amplitude and phase of the output. Although the method requires 500 measurement repetitions at different incidence angles. Mahalati et al. (2013) develop a method in which the number of resolvable image features approached four times the number of spatial modes. Papadopoulos et al. (2012) develop a digital phase conjugation technique to restore images without the requirement of calculating the full TM. Despite this, the solution of finding the TM, or part of it, has to be repeated for every different fibre, for every different length, under each different bending scenario, and for each different temperature, reducing the practicality of this solution. Plöschner et al. (2015) developed a procedure that could also incorporate bending through a precise characterisation of the fibre and a theoretical model.

Another approach to inverting the transmission effects is to use machine learning to solve the inverse problem. Moran et al. (2018) and Caramazza et al. (2019) approached solving the inverse problem through the use of either a Real and Complex linear model with a Hadamard layer to model the power drop off of the spatial light modulator. At its core this approach requires the use of a linear model, which maps from speckled images to the original images and scales as $\mathcal{O}(N_s^2 N_o^2)$, where

$N_s$ is the resolution of the speckled image and $N_o$ is the resolution of the original image. This is a very memory-expensive operation and restricts the scalability of the approach to relatively low resolution images. Further, Borhani et al. (2018) use a convolutional U-Net model to invert the transmission effects, which although using spatially localised convolutions is more memory efficient than a linear model, the speckled images are not in a similar spatial arrangement. An indication of this is likely in the fact their model requires 14 hidden layers to learn the inversion of $32 \times 32$ resolution images. Fan et al. (2019) use a convolutional neural network to invert transmission effects, where the convolutions and pooling reduce the speckled resolution down before a dense linear layer predicts the original image. As a result the model will be sensitive to the learned down-sampling of the convolutions and suffer from similar scalability issues to Moran et al. (2018) due to the inclusion of a dense linear layer. Further, Rahmani et al. (2018) also use a convolutional model and despite not having a dense linear layer, the model does comprise 22 convolutional layers, so, for low resolution, $28 \times 28$, images it is not surprising the model can overcome the mapping between two image domains with non-similar spatial arrangements as a mapping from each speckled pixel to each original image pixel is possible. There is therefore no evidence that the convolutional based models truly improve the scalability issue as these models are very large considering the resolution of images. Finally, of the machine learning based methods for inverting the transmission effects Rahmani et al. (2018) make the first attempt to demonstrate generalisation outside of the training domain, although the results demonstrate limited evidence of true generalisation due to the choice of images featuring limited high frequency features and low quality reconstructions, while Caramazza et al. (2019) demonstrate stronger generalisation due to the testing data choice, but there is still scope to improve the reconstruction quality outside of the training domain.

### A.3 Training Details

Throughout the paper we make use of three different datasets that are commonly used in machine learning, namely MNIST (LeCun et al., 1998), fMNIST (Xiao et al., 2017), and ImageNet (Deng et al., 2009). Where MNIST and fMNIST are also commonly used in the study of inverting transmission effects of MMFs due to their low resolutions. MNIST images are $28 \times 28$ images of handwritten digits between 0 and 9. fMNIST images are $28 \times 28$ images of items of clothing, such as trousers, jumpers, or shoes. ImageNet is a large scale dataset of higher resolution images which we size to be $256x256$. We only use a subset of the imageNet dataset to demonstrate the ability to invert such high resolution images. For each of these datasets we split into a training and testing dataset. The training dataset is used for training, while the testing dataset is not used during training and we use this to test the model after training. We use the testing dataset to generate all the predicted images throughout the paper.

We train each model for 200 epochs with the stochastic gradient descent algorithm. We use the mean squared error as a loss function between the original images and the predicted images. We do not use any regularisation on the model weights. For the Real and Complex linear models and the Bessel equivariant model we train each model for 200 epochs. For the post-processing model, as it is an addition to the Bessel equivariant model, we train this model for 200 epochs after the Bessel equivariant model has already been trained for 200 epochs and the model weights frozen. For the training, all models are trained on 1 Titan RTX GPU with 24GiB of memory. We did attempt training the complex linear model on a A6000 GPU with 48GB of memory on ImageNet to check if this was possible, but it required more memory.

We utilise two different sources of data throughout the experimental section. The first of these is collected using a theoretical TM for which we have both the amplitude and phase information for the speckled patterns. In addition, we also have access to the original and inverted images, where the inverted images are created by first passing an image through the TM and then back through its inverse. The second source of data is that collected by Moran et al. (2018), which is data collected in the lab using a real fibre. For this data we only have the amplitude information for the speckled patterns. In addition, we also have the original images which have constant phase. This data has no licence and it accessible on the projects GitHub page.

## A.4 Invertibility of TMs

Given a TM, which models how an image propagates through a fiber, it is possible to invert the matrix and get the corresponding mapping back from the speckled image space to the original image space. We provide details of the construction of TMs in Appendix 2.2. Utilising this we can inspect information loss due to the limited number of modes of the fibre by passing the image through the TM and back through the inverse to create an inverted image. Therefore, if we knew the TM this inverted image would be the recoverable information. We show an example of the original image, the speckled image, and the inverted image in Figure 9.

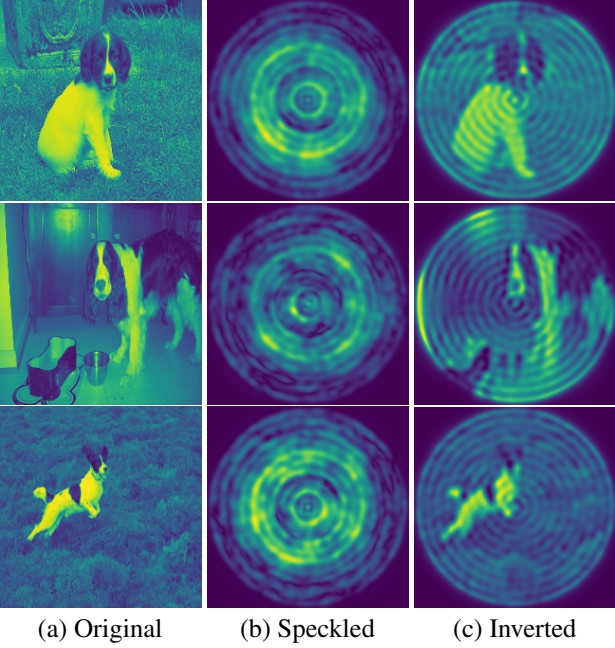

    (a) Original      (b) Speckled      (c) Inverted

Figure 9: (a) The original image. (b) The speckled image created by passing the original images through a theoretical TM. (c) The inverted image created by passing the speckled images through the inverted theoretical TM.

This is useful as it allows us to separate the task of inverting the transmission effects of a fibre into that which could be reconstructed by understanding the physics of the TM and that which requires generating due to being lost information. We believe this could therefore be viewed as two tasks: (1) a task of inverting the transmission effects which would have the aim of generating the inverted images and (2) a task, similar to a super-resolution task, of predicting the original images from the output of the first task.

## A.5 Results for Accounting for Losses in a Real System

As we noted in Section 4.1 the assumption that the mapping between Bessel bases is diagonal holds in theory, although in practice an imperfect system could lead to a necessity to relax this assumption. Here we explore this for the MNIST data collected using a real fibre by building three different versions of our model (1) the strong assumption of a diagonal matrix, (2) relaxing the assumption to allow five elements either side of the diagonal to be populated, (3) relaxing the assumption to allow ten elements either side of the diagonal to be populated, and (4) one with a full matrix mapping between bases. If the fibre and experimental set-up was in an ideal setting we would expect model (1) to produce results as strong as those demonstrated in Section 4.2 where data was created with a theoretical TM. On the other hand, model (2) allowing 5 elements to be populated either side of the diagonal would allow the model to capture some of the block diagonal structure seen in Carpenter et al. (2014), although this would still occlude the mapping between modes further apart in our mapping space. Similarly, (3) allows for capturing more of the block diagonal structure by allowing 10 elements to be populated either side of the diagonal. Finally, model (4) allows the greatest flexibility and has the potential to capture deviations from theory as seen in practice, although, this

model does not take advantage of the known diagonal structure that this mapping function has and is therefore over-parameterised.

Figure 10 demonstrates that as the diagonal assumption of the mapping function between Bessel bases is relaxed, our Bessel equivariant model produced images closer to that original image through capturing more high frequency detail and predicting less noise. Our model with five elements either side of the diagonal populated with trainable parameters achieves a comparable result to the Real and Complex linear models, and our most constrained model is able to generate digits that like the correct digit. This is a promising result as it demonstrates that our model allows for a design choice of complexity, with the option to trade off performance for a reduced memory requirement whilst maintaining accurate results even in the lowest memory configuration; this is something that is not possible with the Real or Complex linear models. Choosing a model with a low number of off-diagonal trainable elements, we believe, is the best trade off between considering losses and imperfections of the real-world while benefiting from the known sparsity of the mapping function between Bessel bases.

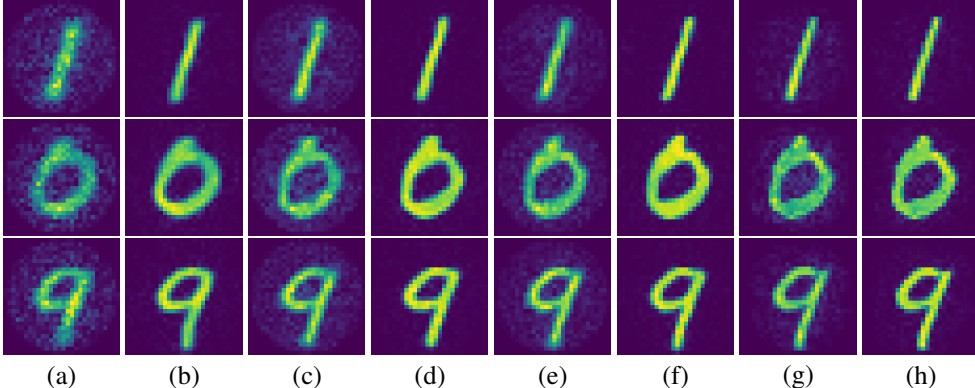

| (a) | (b) | (c) | (d) | (e) | (f) | (g) | (h) |

Figure 10: Comparison of predicted images from inverting transmission effects of a MMF with varying relaxations on the assumption the fibre has a diagonal fibre propagation matrix. (a) the output of the Bessel equivariant model with a diagonal propagation matrix assumption between Bessel bases, (b) the output of the combination of Bessel equivariant with a diagonal propagation matrix assumption between Bessel bases and post-processing model, (c) the output of the Bessel equivariant model with 5 diagonal offsets in the propagation matrix between Bessel bases, (d) the output of the Bessel equivariant model with 5 diagonal offsets in the propagation matrix between Bessel bases and post-processing model, (e) the output of the Bessel equivariant model with 10 diagonal offsets in the propagation matrix between Bessel bases, (f) the output of the Bessel equivariant model with 10 diagonal offsets in the propagation matrix between Bessel bases and post-processing model, (g) the output of the Bessel equivariant model with a full propagation matrix between Bessel bases, and (h) the output of the Bessel equivariant model with a full propagation matrix between Bessel bases and post-processing model.

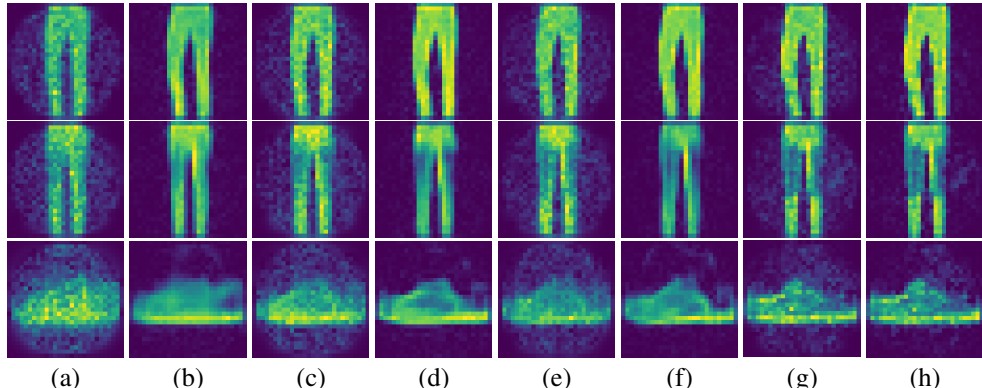

| (a) | (b) | (c) | (d) | (e) | (f) | (g) | (h) |

Figure 11: Comparison of predicted images from inverting transmission effects of a MMF with varying relaxations on the assumption the fibre has a diagonal fibre propagation matrix. (a) the output of the Bessel equivariant model with a diagonal propagation matrix assumption between Bessel bases, (b) the output of the combination of Bessel equivariant with a diagonal propagation matrix assumption between Bessel bases and post-processing model, (c) the output of the Bessel equivariant model with 5 diagonal offsets in the propagation matrix between Bessel bases, (d) the output of the Bessel equivariant model with 5 diagonal offsets in the propagation matrix between Bessel bases and post-processing model, (e) the output of the Bessel equivariant model with 10 diagonal offsets in the propagation matrix between Bessel bases, (f) the output of the Bessel equivariant model with 10 diagonal offsets in the propagation matrix between Bessel bases and post-processing model, (g) the output of the Bessel equivariant model with a full propagation matrix between Bessel bases, and (h) the output of the Bessel equivariant model with a full propagation matrix between Bessel bases and post-processing model.

## A.6 Effects of Noise on Speckled Images

Here we compare the effect of noise on the speckled images when using a theoretical TM. In each case the speckled images are saturated at $90\%$ of there maximum value and Gaussian noise is added with different standard deviation values. We demonstrate the reconstruction ability of our model in Figure 12 when Gaussian noise with $0.01$ standard deviation, in Figure 13 when Gaussian noise with $0.05$ standard deviation, in Figure 14 when Gaussian noise with $0.1$ standard deviation, and in Figure 15 when Gaussian noise with $0.5$ standard deviation. Further, we provide the loss values in Table 5. This demonstrates that our approach is robust to noise.

Table 5: Comparison of the loss values of each model trained with F-MNIST data when the speckled images are saturated at $90\%$ and Gaussian noise is added with standard deviation given in the column Noise Level.

| Noise Level | Model | Train Loss | Test Loss |
|---|---|---|---|
| 0.01 | Bessel Equivariant | 0.0140 | 0.0142 |
| | Bessel Equivariant + Post Proc | 0.0033 | 0.0033 |
| 0.05 | Bessel Equivariant | 0.0147 | 0.0149 |
| | Bessel Equivariant + Post Proc | 0.0043 | 0.0043 |
| 0.1 | Bessel Equivariant | 0.0166 | 0.0168 |
| | Bessel Equivariant + Post Proc | 0.0049 | 0.0049 |
| 0.5 | Bessel Equivariant | 0.0373 | 0.0377 |
| | Bessel Equivariant + Post Proc | 0.0164 | 0.0166 |

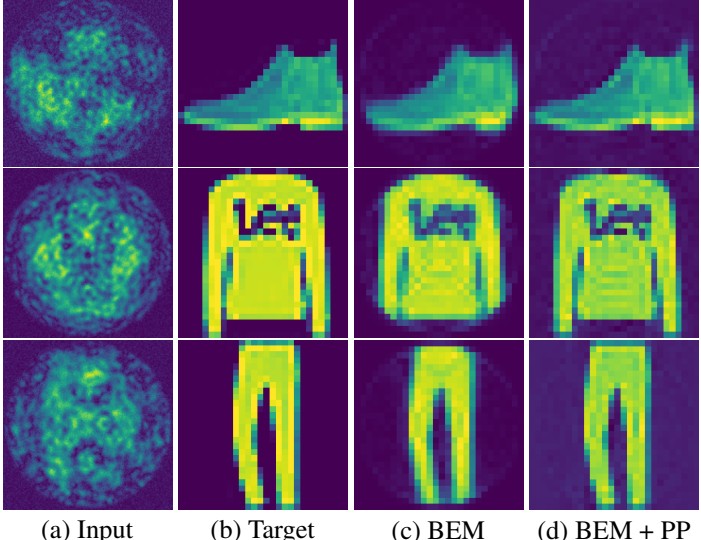

|(a) Input|(b) Target|(c) BEM|(d) BEM + PP|

Figure 12: Comparison of predicted images from inverting transmission effects of a MMF using FMnist data created with a theoretical TM when the speckled images are saturated at 90% of their maximum value and Gaussian noise with 0.01 standard deviation added. (a) The input noisy speckled image, (b) the target original image to reconstruct, (c) the output of the Bessel equivariant model, and (d) the output of the combination of Bessel equivariant and post-processing model.

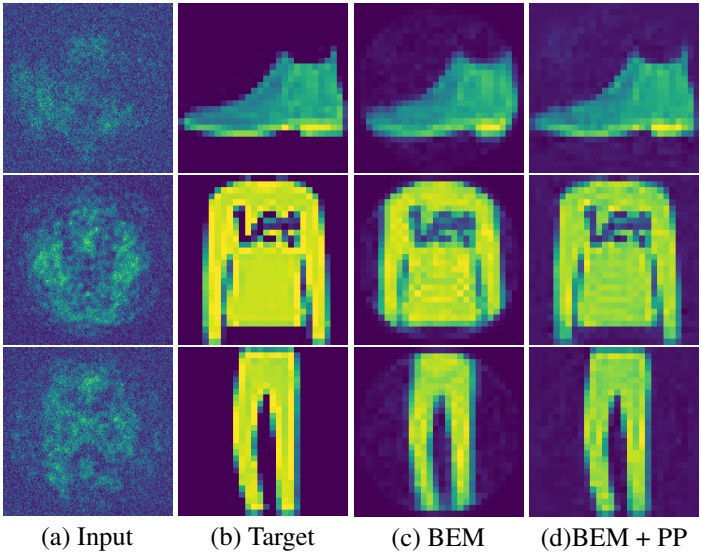

|(a) Input|(b) Target|(c) BEM|(d)BEM + PP|

Figure 13: Comparison of predicted images from inverting transmission effects of a MMF using FMnist data created with a theoretical TM when the speckled images are saturated at 90% of their maximum value and Gaussian noise with 0.05 standard deviation added. (a) The input noisy speckled image, (b) the target original image to reconstruct, (c) the output of the Bessel equivariant model, and (d) the output of the combination of Bessel equivariant and post-processing model.

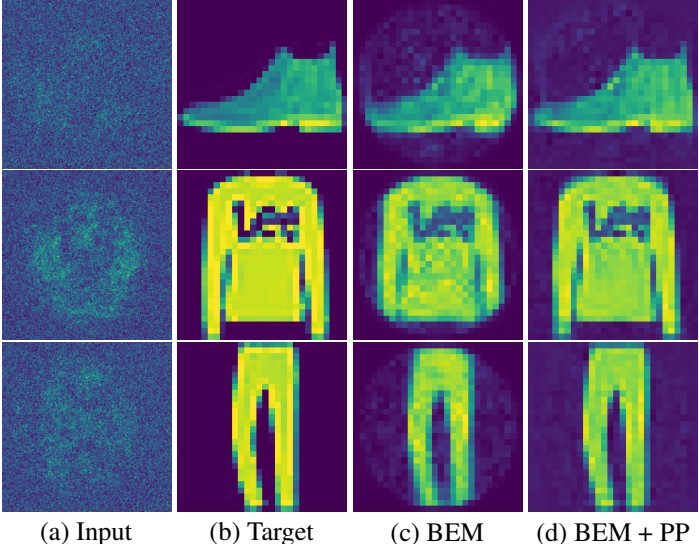

|              |              |              |              |
|:------------:|:------------:|:------------:|:------------:|
| (a) Input    | (b) Target   | (c) BEM      | (d) BEM + PP |

Figure 14: Comparison of predicted images from inverting transmission effects of a MMF using FMnist data created with a theoretical TM when the speckled images are saturated at $90\%$ of their maximum value and Gaussian noise with $0.1$ standard deviation added. (a) The input noisy speckled image, (b) the target original image to reconstruct, (c) the output of the Bessel equivariant model, and (d) the output of the combination of Bessel equivariant and post-processing model.

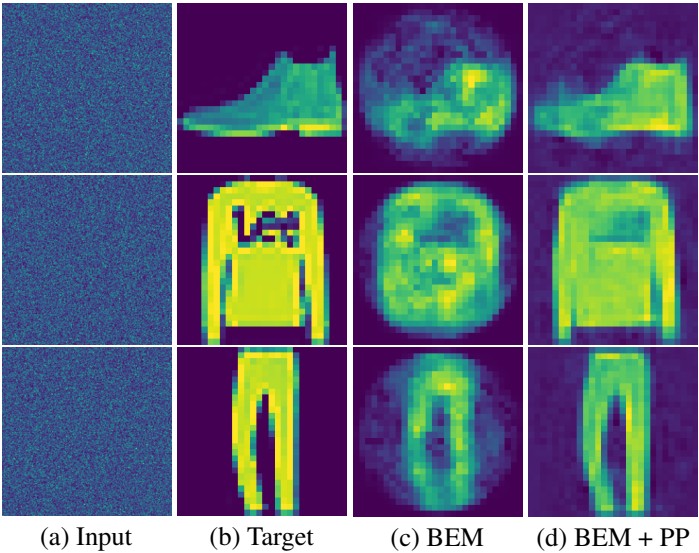

|              |              |              |              |
|:------------:|:------------:|:------------:|:------------:|
| (a) Input    | (b) Target   | (c) BEM      | (d) BEM + PP |

Figure 15: Comparison of predicted images from inverting transmission effects of a MMF using FMnist data created with a theoretical TM when the speckled images are saturated at $90\%$ of their maximum value and Gaussian noise with $0.05$ standard deviation added. (a) The input noisy speckled image, (b) the target original image to reconstruct, (c) the output of the Bessel equivariant model, and (d) the output of the combination of Bessel equivariant and post-processing model.

## A.7 Additional Reconstructions - Real Fibre MNIST

Table 6: Comparison of the loss values of each model trained with MNIST data.

| Model | MNIST Train Loss | Test Loss |
|---|---|---|
| Real Linear | 0.00466 | 0.00683 |
| Complex Linear | 0.00396 | 0.00684 |
| Bessel Equivariant Diag | 0.03004 | 0.03125 |
| Bessel Equivariant Diag + Post Proc | 0.01317 | 0.01453 |
| Bessel Equivariant 5 Off Diag | 0.01782 | 0.01931 |
| Bessel Equivariant 5 Off Diag + Post Proc | 0.00658 | 0.00740 |
| Bessel Equivariant 10 Off Diag | 0.01362 | 0.01521 |
| Bessel Equivariant 10 Off Diag + Post Proc | 0.00488 | 0.00576 |
| Bessel Equivariant Full | 0.00300 | 0.00684 |
| Bessel Equivariant Full + Post Proc | **0.00145** | **0.00378** |

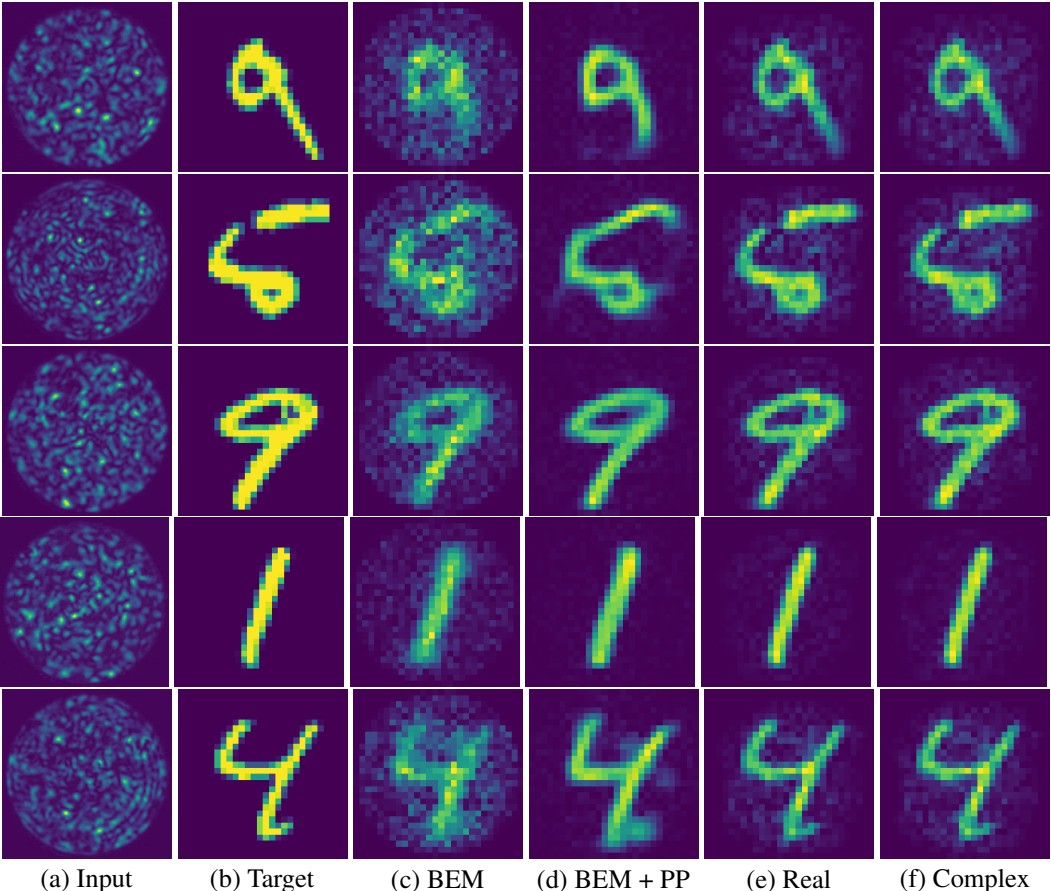

(a) Input    (b) Target    (c) BEM    (d) BEM + PP    (e) Real    (f) Complex

Figure 16: Comparison of predicted images from inverting transmission effects of a MMF. (a) The input speckled image, (b) the target original image to reconstruct, (c) the output of the Bessel equivariant model, (d) the output of the combination of Bessel equivariant and post-processing model, (e) the output of the Real valued linear model, and (f) the output of the Complex valued linear model.

## A.8 Additional Reconstructions - Real Fibre fMNIST

Table 7: Comparison of the loss values of each model trained with fMNIST data.

| | fMNIST | |
| --- | --- | --- |
| Model | Train Loss | Test Loss |
| Real Linear | 0.00586 | 0.01061 |
| Complex Linear | 0.00509 | 0.01061 |
| Bessel Equivariant Diag | 0.02903 | 0.03140 |
| Bessel Equivariant Diag + Post Proc | 0.01609 | 0.01749 |
| Bessel Equivariant 5 Off Diag | 0.02057 | 0.02354 |
| Bessel Equivariant 5 Off Diag + Post Proc | 0.01138 | 0.01315 |
| Bessel Equivariant 10 Off Diag | 0.01788 | 0.02140 |
| Bessel Equivariant 10 Off Diag + Post Proc | 0.00943 | 0.01162 |
| Bessel Equivariant Full | 0.00548 | 0.01380 |
| Bessel Equivariant Full + Post Proc | **0.00306** | **0.00964** |

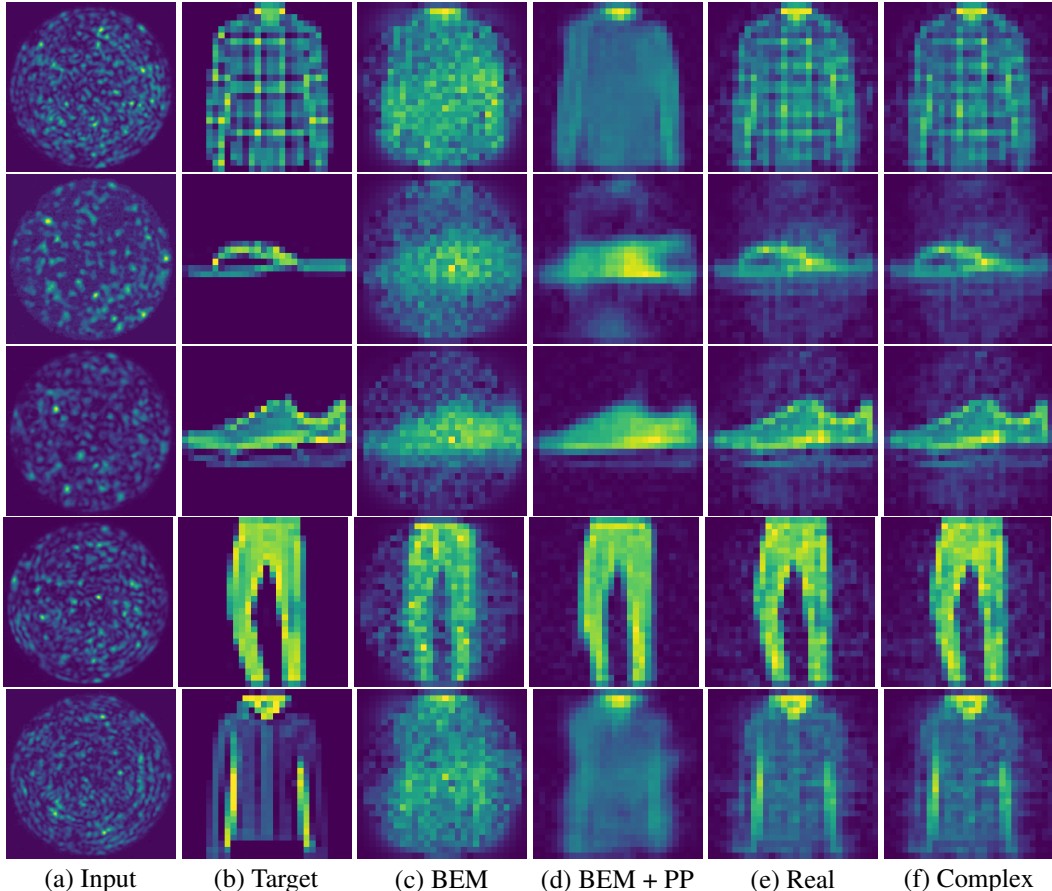

(a) Input  (b) Target  (c) BEM  (d) BEM + PP  (e) Real  (f) Complex

Figure 17: Comparison of predicted images from inverting transmission effects of a MMF. (a) The input speckled image, (b) the target original image to reconstruct, (c) the output of the Bessel equivariant model, (d) the output of the combination of Bessel equivariant and post-processing model, (e) the output of the Real valued linear model, and (f) the output of the Complex valued linear model.

## A.9 Additional Reconstructions - Theory TM fMNIST

Table 8: Comparison of the loss values of each model trained with fMNIST data.

| Model | fMNIST Train Loss | fMNIST Test Loss |
|---|---|---|
| Real Linear | 0.0256 | 0.0250 |
| Complex Linear | 0.0149 | 0.0146 |
| Bessel Equivariant | 0.0141 | 0.0139 |
| Bessel Equivariant + Post Proc | **0.0032** | **0.0032** |

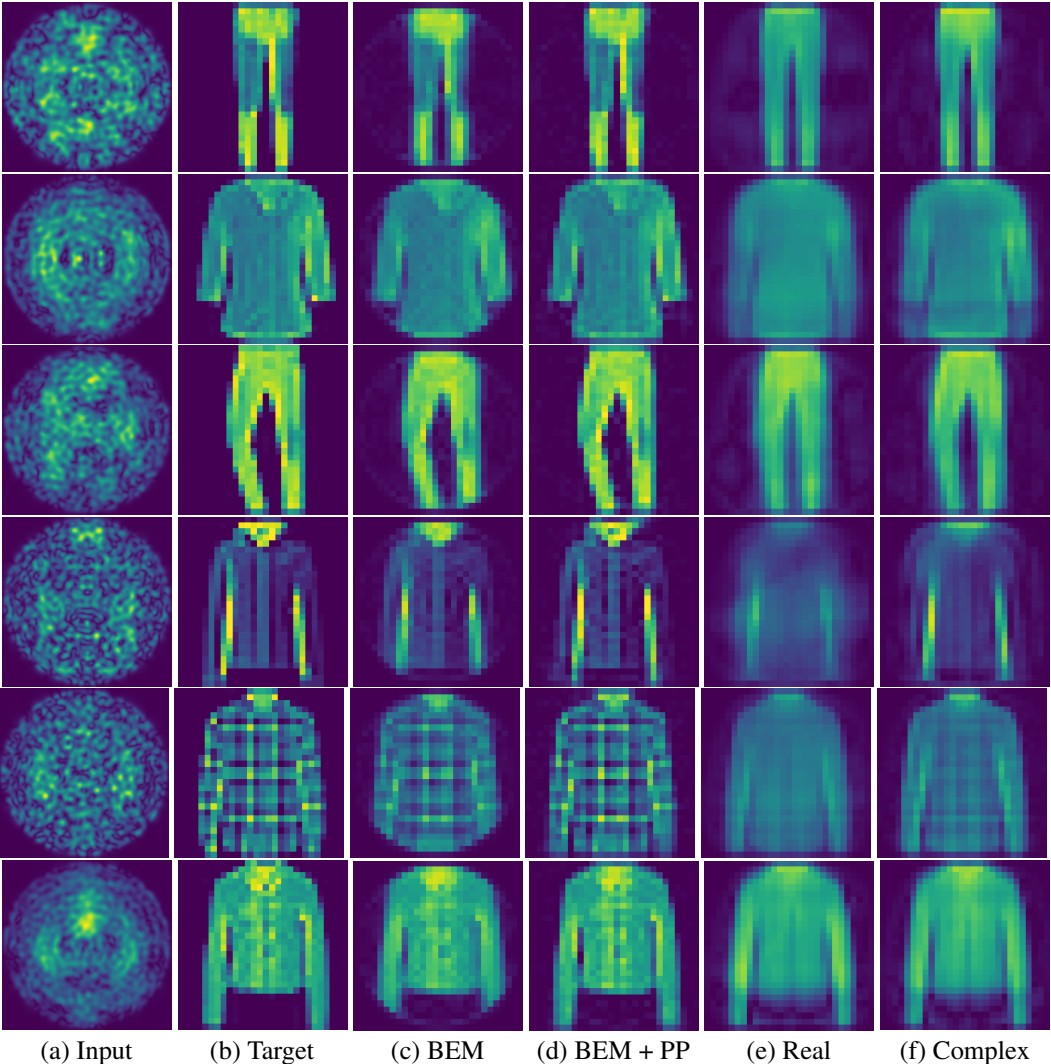

(a) Input    (b) Target    (c) BEM    (d) BEM + PP    (e) Real    (f) Complex

Figure 18: Comparison of predicted images from inverting transmission effects of a MMF. (a) The input speckled image, (b) the target original image to reconstruct, (c) the output of the Bessel equivariant model, (d) the output of the combination of Bessel equivariant and post-processing model, (e) the output of the Real valued linear model, and (f) the output of the Complex valued linear model.

## A.10 Additional Reconstructions - Theory TM MNIST

Table 9: Comparison of the loss values of each model trained with fMNIST data.

| Model | MNIST Test Loss |
|---|---|
| Real Linear | 0.0641 |
| Complex Linear | 0.0363 |
| Bessel Equivariant | **0.0026** |
| Bessel Equivariant + Post Proc | 0.0028 |

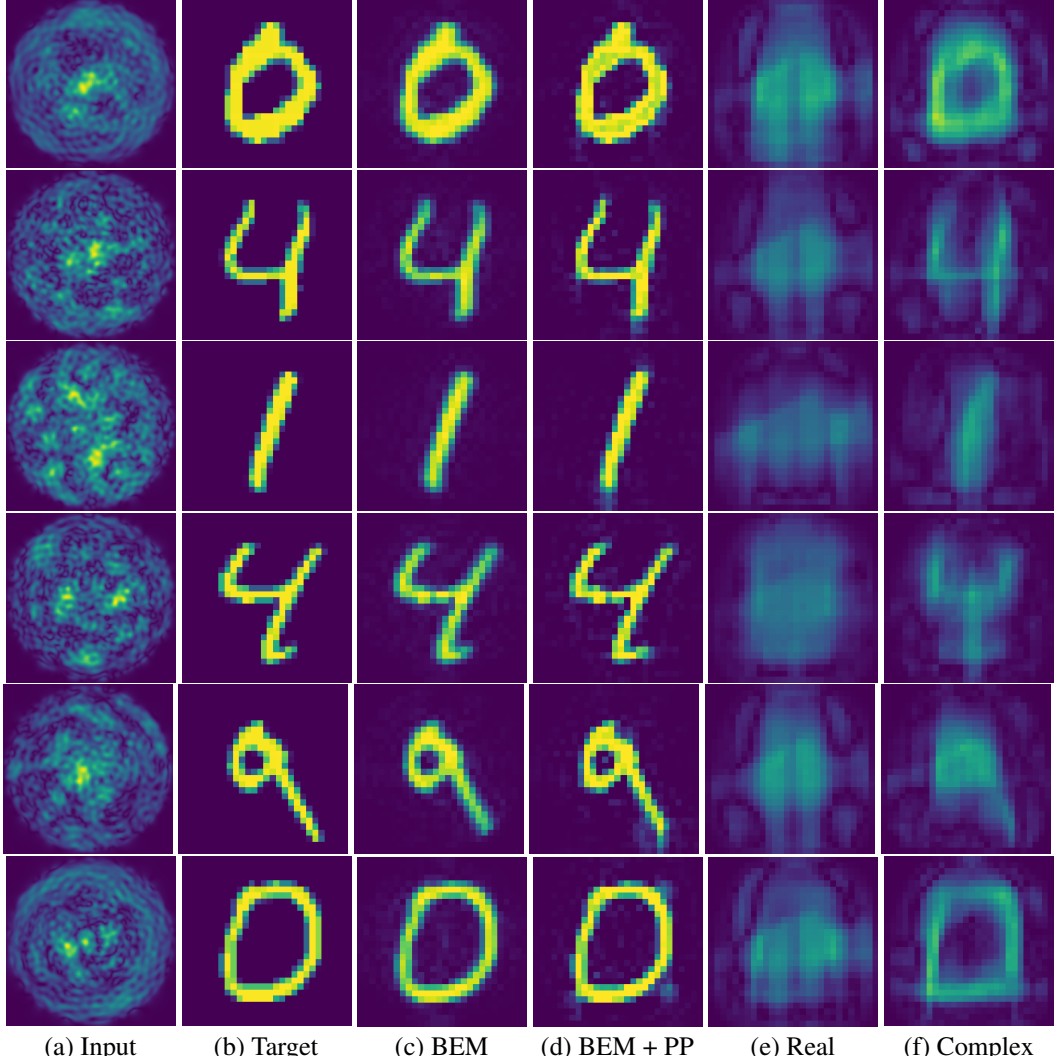

|  |  |  |  |  |  |
|---|---|---|---|---|---|
| (a) Input | (b) Target | (c) BEM | (d) BEM + PP | (e) Real | (f) Complex |

Figure 19: Comparison of predicted images from inverting transmission effects of a MMF. (a) The input speckled image, (b) the target original image to reconstruct, (c) the output of the Bessel equivariant model, (d) the output of the combination of Bessel equivariant and post-processing model, (e) the output of the Real valued linear model, and (f) the output of the Complex valued linear model.

## A.11 Additional Reconstructions - Theory TM ImageNet

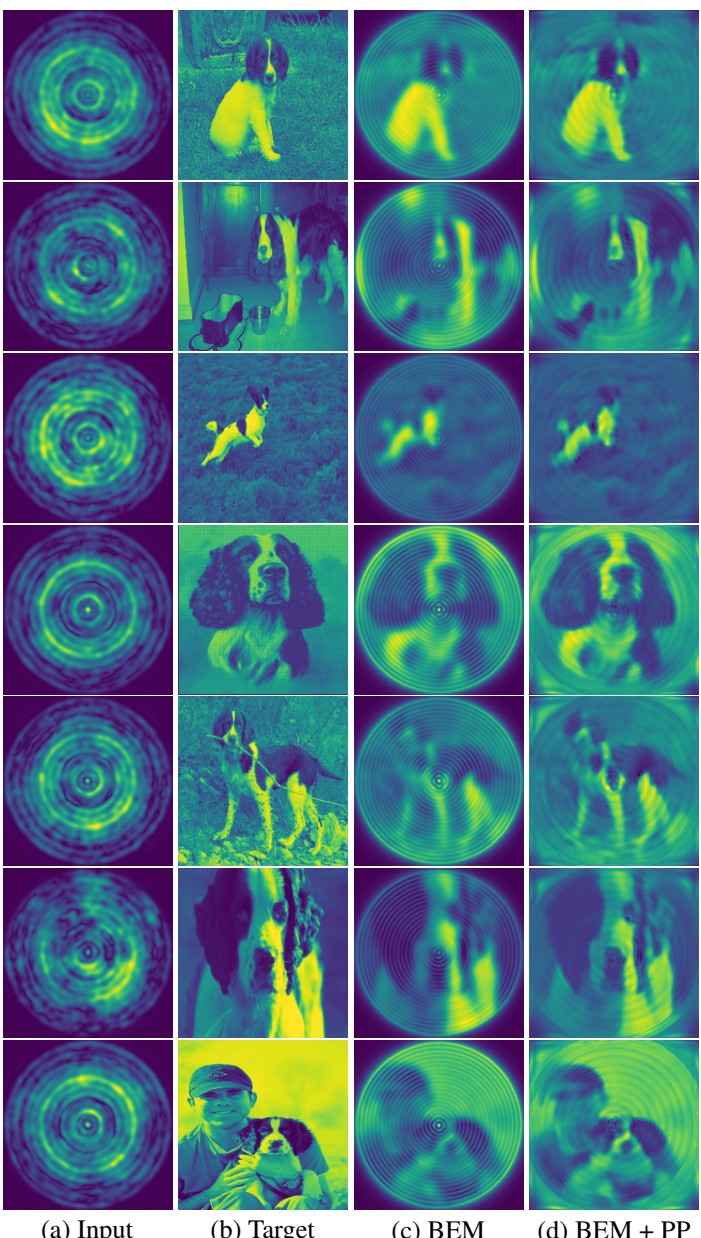

(a) Input     (b) Target     (c) BEM     (d) BEM + PP

Figure 20: Comparison of predicted images from inverting transmission effects of a MMF. (a) The input speckled image, (b) the target original image to reconstruct, (c) the output of the Bessel equivariant model, and (d) the output of the combination of Bessel equivariant and post-processing model.

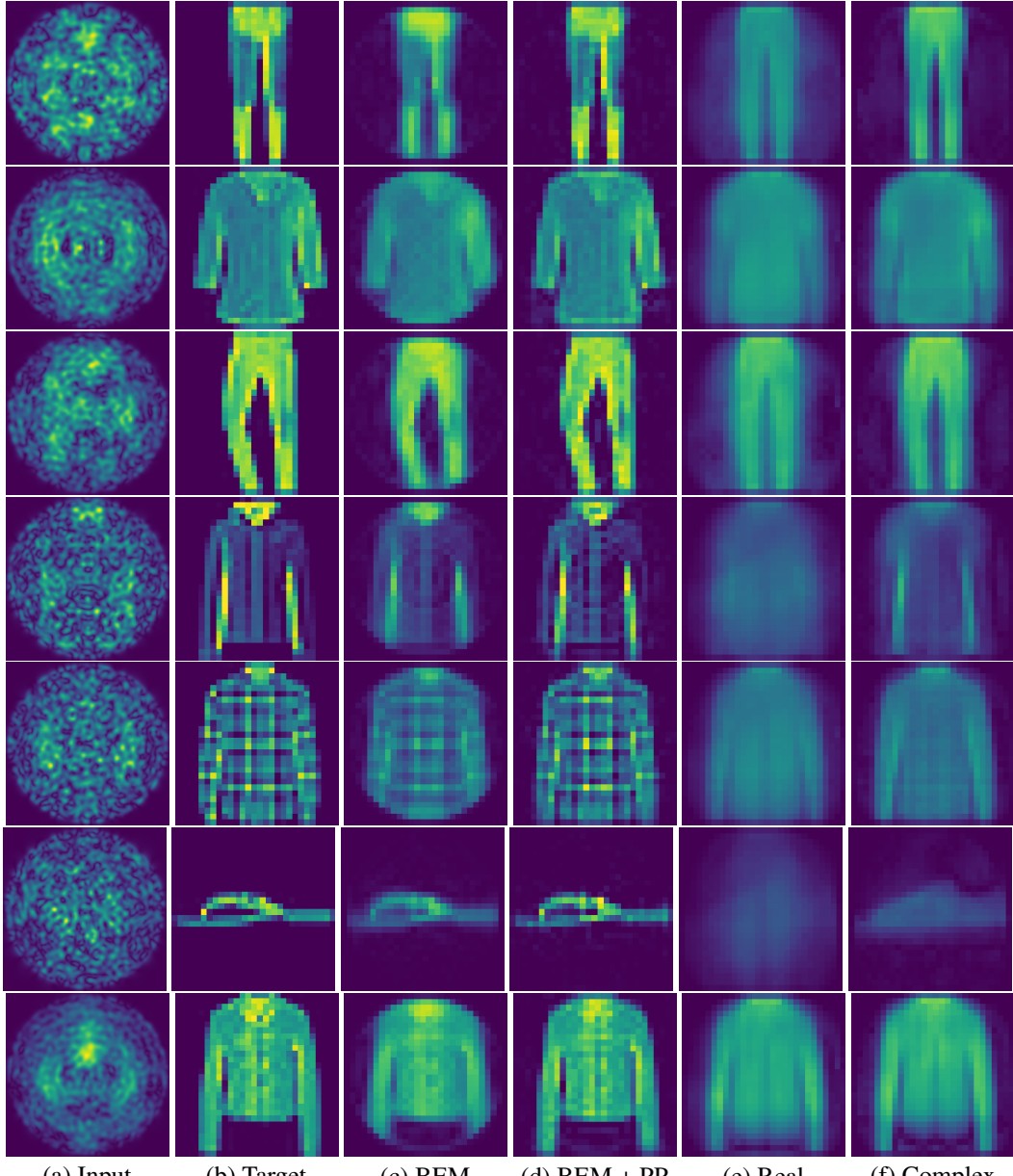

(a) Input        (b) Target        (c) BEM        (d) BEM + PP        (e) Real        (f) Complex

Figure 21: Comparison of predicted images from inverting transmission effects of a MMF. The training dataset was reduced from the original size of 12000 to 6000. (a) The input speckled image, (b) the target original image to reconstruct, (c) the output of the Bessel equivariant model, (d) the output of the combination of Bessel equivariant and post-processing model, (e) the output of the Real valued linear model, and (f) the output of the Complex valued linear model. The data used was fMNIST with speckled patterns created with a theoretical TM.

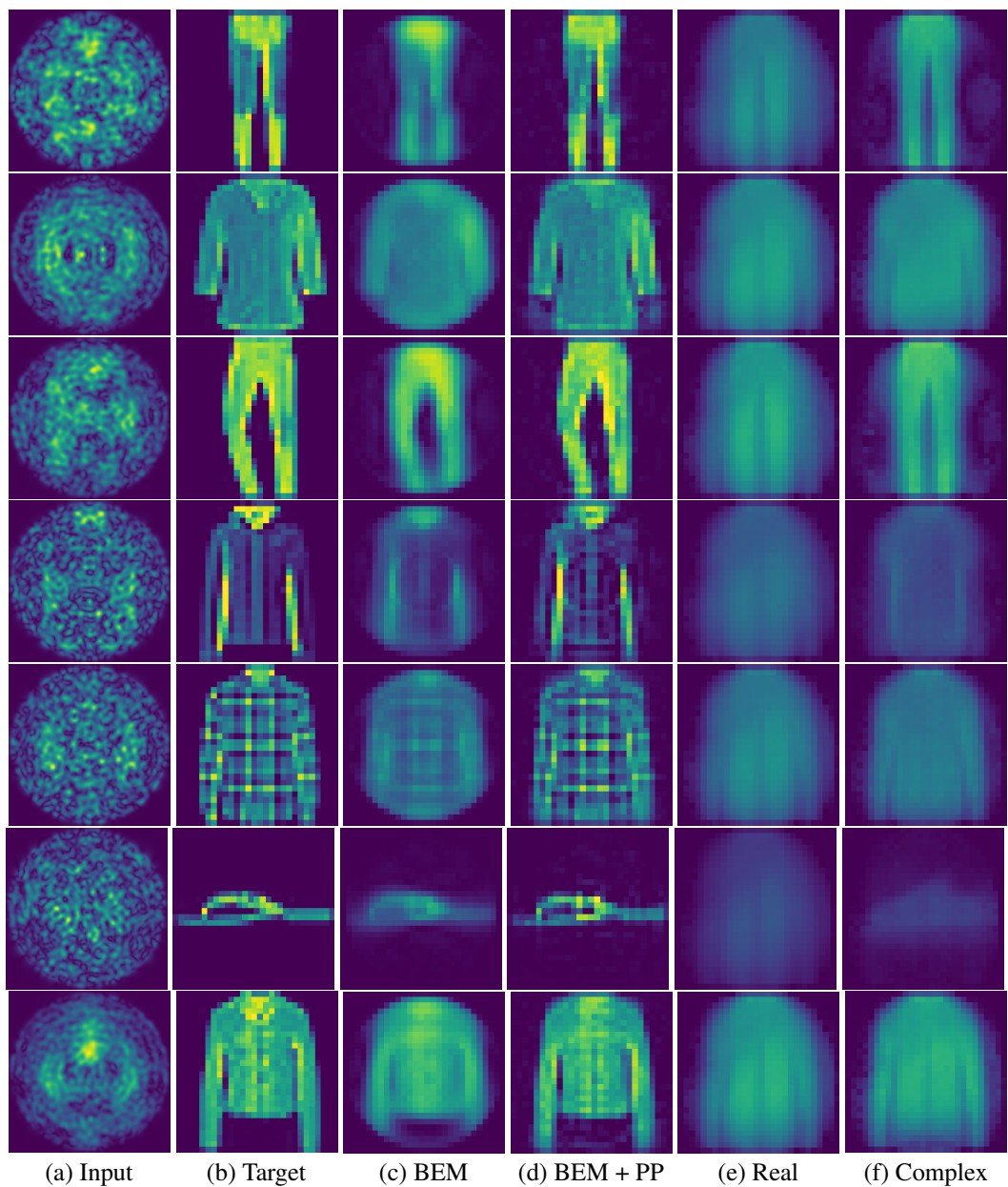

| (a) Input | (b) Target | (c) BEM | (d) BEM + PP | (e) Real | (f) Complex |

Figure 22: Comparison of predicted images from inverting transmission effects of a MMF. The training dataset was reduced from the original size of 12000 to 2400. (a) The input speckled image, (b) the target original image to reconstruct, (c) the output of the Bessel equivariant model, (d) the output of the combination of Bessel equivariant and post-processing model, (e) the output of the Real valued linear model, and (f) the output of the Complex valued linear model. The data used was fMNIST with speckled patterns created with a theoretical TM.

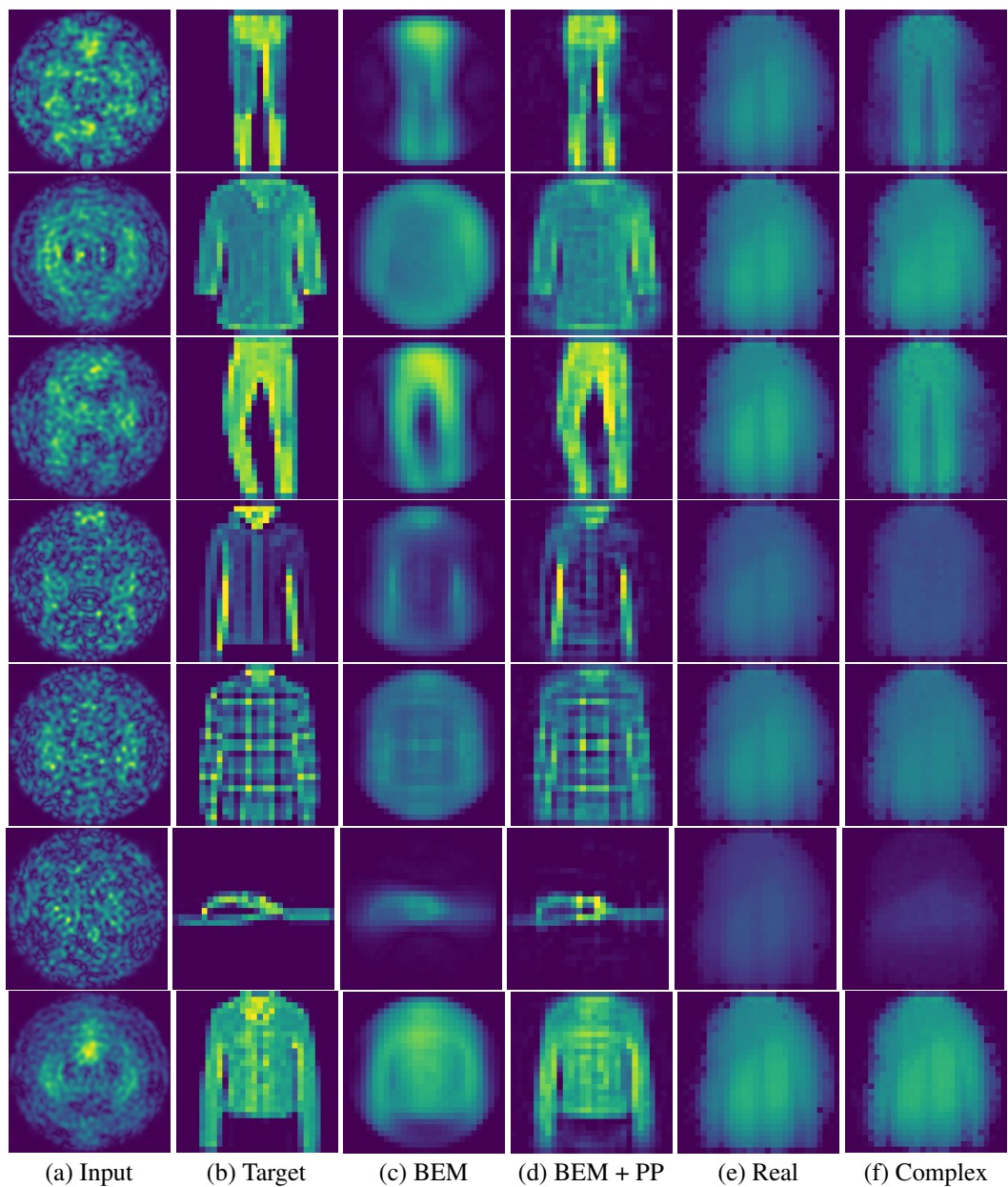

| (a) Input | (b) Target | (c) BEM | (d) BEM + PP | (e) Real | (f) Complex |

Figure 23: Comparison of predicted images from inverting transmission effects of a MMF. The training dataset was reduced from the original size of 12000 to 1200. (a) The input speckled image, (b) the target original image to reconstruct, (c) the output of the Bessel equivariant model, (d) the output of the combination of Bessel equivariant and post-processing model, (e) the output of the Real valued linear model, and (f) the output of the Complex valued linear model. The data used was fMNIST with speckled patterns created with a theoretical TM.

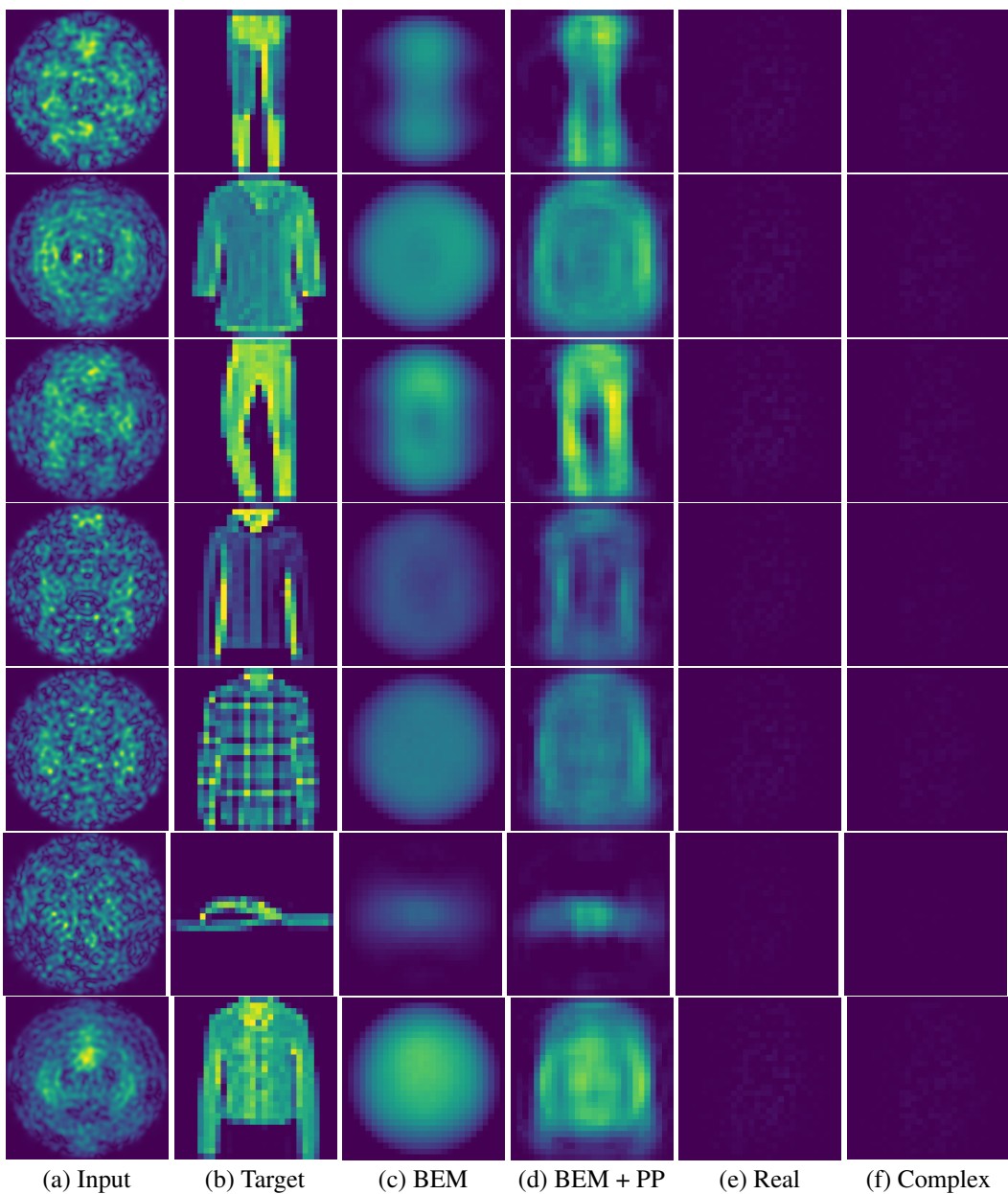

(a) Input    (b) Target    (c) BEM    (d) BEM + PP    (e) Real    (f) Complex

Figure 24: Comparison of predicted images from inverting transmission effects of a MMF. The training dataset was reduced from the original size of 12000 to 120. (a) The input speckled image, (b) the target original image to reconstruct, (c) the output of the Bessel equivariant model, (d) the output of the combination of Bessel equivariant and post-processing model, (e) the output of the Real valued linear model, and (f) the output of the Complex valued linear model. The data used was fMNIST with speckled patterns created with a theoretical TM.

### A.13 Impacts of Under Parameterising the set of Bessel Function Bases - Theory TM fMNIST

Here we consider the impact on under parameterising the Bessel function basis. For this we reduce the number of radial frequencies that feature in the set of bases functions. The original basis set we utilised for the data collected using a theoretical fibre for fMNIST comprises of 21 radial frequencies and 1061 bases. Here we show the original results using this full bases set in Figure 26 column (b). Next we show the results of removing the high frequency bases by only considering the first 14 radial frequencies, which amounts to having 932 bases in Figure 26 column (c). In addition, we show the results of removing further high frequency bases by only considering the first 7 radial frequencies, which amounts to having 567 bases in Figure 26 column (d). Finally, we show the results of removing further high frequency bases by only considering the first 4 radial frequencies, which amounts to having 322 bases in Figure 26 column (e).

Table 10: Comparison of the loss values of each model trained with fMNIST data created with a theoretical TM.This shows the impact of under parameterising the Bessel function basis set by removing higher frequency modes. 21 radial frequencies is the natural choice for the fibre considered, hence all values less than 21 give an under parameterised model.

| Model | Radial Frequencies | # Modes | Train Loss | Test Loss |
|---|---|---|---|---|
| Bessel Equivariant | 21 | 1061 | 0.0141 | 0.0139 |
| Bessel Equivariant + Post Proc | 21 | 1061 | **0.0032** | **0.0032** |
| Bessel Equivariant | 14 | 932 | 0.0158 | 0.0156 |
| Bessel Equivariant + Post Proc | 14 | 932 | 0.0036 | 0.0036 |
| Bessel Equivariant | 7 | 567 | 0.0203 | 0.0201 |
| Bessel Equivariant + Post Proc | 7 | 567 | 0.0053 | 0.0053 |
| Bessel Equivariant | 4 | 332 | 0.0299 | 0.0296 |
| Bessel Equivariant + Post Proc | 4 | 332 | 0.0090 | 0.0090 |

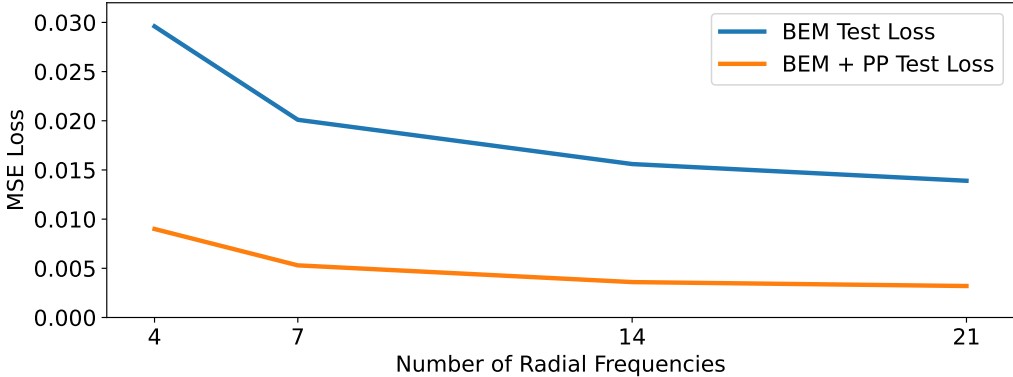

Figure 25: Comparison of the loss values of each model trained with fMNIST data created with a theoretical TM. This shows the impact of under parameterising the Bessel function basis set by removing higher frequency modes. 21 radial frequencies is the natural choice for the fibre considered, hence all values less than 21 give an under parameterised model.

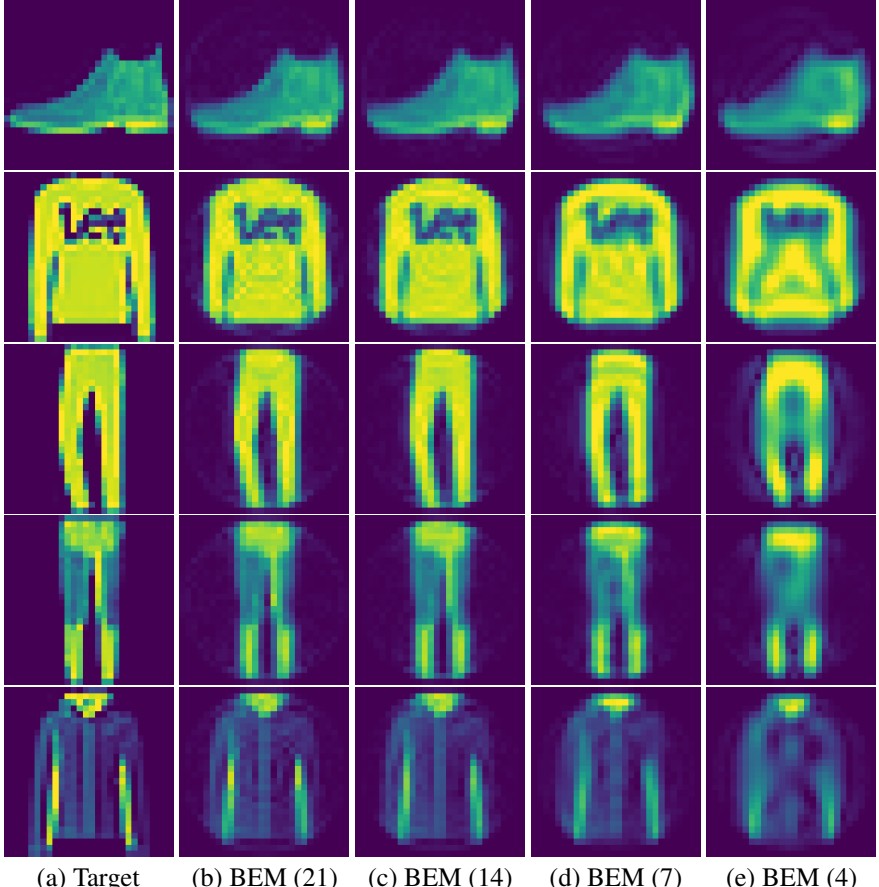

|          (a) Target          (b) BEM (21)          (c) BEM (14)          (d) BEM (7)          (e) BEM (4)          |

Figure 26: Comparison of predicted images from inverting transmission effects of a MMF, when reducing the number of radial frequencies present in the Bessel function basis. (a) The input speckled image. (b) The number of radial frequencies present in the Bessel function baiis was left at the original value of 21, which represents 1061 bases. (c) The number of radial frequencies present in the Bessel function basis was reduced from the original value of 21 to 14, which represents a reduction in the number of bases from the original value of 1061 to 930. (d) The number of radial frequencies present in the Bessel function basis was reduced from the original value of 21 to 7, which represents a reduction in the number of bases from the original value of 1061 to 567. (e) The number of radial frequencies present in the Bessel function basis was reduced from the original value of 21 to 4, which represents a reduction in the number of bases from the original value of 1061 to 322. The data used was fMNIST with speckled patterns created with a theoretical TM.

## A.14 Negative Societal Impacts

Our work enables imaging using a multi-mode optical fibre at higher resolutions than has been previously been achievable with a machine learning based approach. In addition, our new method generalises better to new data classes out of the training data classes. Therefore, it better enables imaging using multi-mode fibres to be used practically. Thus, this technology could potentially be used in a way that negatively impacts society through any negative use case of multi-mode optical fibre imaging (e.g. possible uses in espionage). Never-the-less the technology developed during this paper does not have a direct negative impact.