# OpenReview forum: "Bessel Equivariant Networks for Inversion of Transmission Effects in Multi-Mode Optical Fibres"
_NeurIPS.cc/2022/Conference — NeurIPS 2022 Accept_

### Official Review · Reviewer_88iJ · 2022-07-10

**Rating:** 8
**Confidence:** 4
**Soundness:** 4 excellent
**Presentation:** 4 excellent
**Contribution:** 3 good

**Summary:**

The paper reports on a novel NN architecture for image transmission through a multimode fiber. The approach is physics based, and rely on modeling a NN using bessel modes, that mimicks the pre-existing modes in a MMF. the main advantage is to reduce strongly the number of parameters to train (compared to a complex-valued fully connected layer, the current SoTA) and  the required number of examples, while maintaining similar performances and generalization ability. The concept is validated numerically and experimentally, at scale (256*256) previously unattainable.

**Questions:**

-Modes of MMF are usually denoted LP modes in the literature, can the authors comments if they are the same as their Bessels? is it a different terminology, or is there a real difference?


**Limitations:**

I think it really doesn't apply here.

**Strengths And Weaknesses:**

The paper is overall clearly written, the results are sound, and the comparison to the previous litterature is qualitatively and quantitatively appropriate. The main advantage of the technique (less parameters) meaning less memory usage and less examples needed, means that it can scale to complex and large scale images, which is demonstrated.

A minor criticism is that the authors could have cited a recent paper (Cite: Li, Shuhui, et al. "Compressively sampling the optical transmission matrix of a multimode fibre." Light: Science & Applications 10.1 (2021): 1-15.) that also exploit the sparsity of the transmission matrix of a multimode fiber.  The paper in my opinion should be cited and properly put in perspective with their result.

---

> ### Author Response · Authors · 2022-08-01
> **Response to Reviewer**
>
> Thank you for your review and assessment of the strengths of our approach. The assessment of these strengths and the understanding of the model is correct. A further advantage of our approach is that it also generalizes better to new datasets, see Fig 5.
>
> We will address the concerns below:
>
> * A minor criticism is that the authors could have cited a recent paper (Cite: Li, Shuhui, et al. "Compressively sampling the optical transmission matrix of a multimode fibre." Light: Science & Applications 10.1 (2021): 1-15.) that also exploit the sparsity of the transmission matrix of a multimode fiber. The paper in my opinion should be cited and properly put in perspective with their result.
>
> Thank you for pointing out this recently published paper. We have added in the citation for this paper and positioned it with respect to our work within our introduction, noting the exploitation of sparsity in the paper.
>
>
> * Modes of MMF are usually denoted LP modes in the literature, can the authors comments if they are the same as their Bessels? is it a different terminology, or is there a real difference?
>
> This is just a difference in terminology, where we are trying to make the paper more accessible to a machine learning audience. For the LP modes one assumes linearly polarised light and solves the Helmholz equation with appropriate cylindrical boundary conditions to arrive at the spatial fields defined by Bessel functions. Some examples of which are given in Appendix A.4. We have added that these are LP modes in Figure 8 to make this clear.

---

> > ### Comment · Reviewer_88iJ · 2022-08-08
> > **thanks.**
> >
> > I thank the authors for their answers, my points are addressed.

---

### Official Review · Reviewer_iftp · 2022-07-11

**Rating:** 5
**Confidence:** 5
**Soundness:** 3 good
**Presentation:** 3 good
**Contribution:** 3 good

**Summary:**

This paper assumes that the transmission matrices (TMs) associated with multimodal fibers (MMF) are diagonizable using a Bessel Function Basis. Using this assumption, the authors propose a new learning-based technique for imaging through MMFs. The speckle image is first multiplied by the learned diagonal inverse TM and is then denoised by a post-processing CNN. The proposed method is tested on low-resolution toy (MNIST and FMNIST) experimentally captured data from (Mora et al. 2018). It is also tested on simulated TM data on higher-res image-net data.

**Questions:**

How would fitting a low rank approximation of the TM (or its inverse) compare to the proposed method?

**Limitations:**

Not discussed. Method is specific to a particular MMF. Moreover, if that fiber bends (like it would in endoscopy) that fibers TM is likely to change.

**Strengths And Weaknesses:**

## Strengths

The diagonal assumption massively reduced the number of parameters that need to be learned and could make imaging through MMF much easier.

## Weaknesses

The paper makes extremely strong assumptions on the structure of the TM which are not adequately justified. The only experimental validation is on (F)MNIST which isn't convincing. The paper provides no evidence that Bessel functions accurately describe the structure of higher-dimensional TMs. The simulated speckle data (Fig 6) looks considerably more structured than the experimental data (Fig 5), suggesting the adopted model is inaccurate.

Some details are missing. It seems the paper is considering only "incoherent" fiber bundles.

There's no evidence the proposed method generalizes across MMFs.

There are minor inaccuracies in the paper: Line 270 states reducing the calibration dataset size minimizes how long cells have to be imaged; cells are involved in the calibration of a TM.

---

> ### Author Response · Authors · 2022-08-01
> **Response to Reviewer (1/3)**
>
> Thank you for your comment acknowledging the significantly reduced parameter count required to be learned by our model and how this could make imaging through a MMF easier. As we demonstrate in the paper, a further benefit is that our proposed architecture also makes it possible to image with higher-resolution images, which was not possible with previous dense methods. In addition, as we demonstrate in Fig 5 the reduced parameter count required to be learned by our model also improves generalization to new unseen datasets, which is a further benefit.
>
> We will address the concerns below:
>
> * The paper makes extremely strong assumptions on the structure of the TM which are not adequately justified.
> * The paper provides no evidence that Bessel functions accurately describe the structure of higher-dimensional TMs.
>
> We provide extensive details on the generation of our theoretical TMs in Appendix A4;  this includes details about the fundamental physics of optical fibers, demonstrating that Bessel functions describe the structure of the modes of TMs. Furthermore, we provide details about TMs and make a connection to the group theoretic way of thinking used in equivariant neural networks in Appendix A8.
>
> Both A4 and A8 provide known details about optical fibers and the equations governing light propagation. These sections provide the connection between the literature on light propagation through optical fibers and our model, which was perhaps missed because of the appendix being uploaded with the supplementary data. We have re-uploaded our paper, with the main text and appendix in a single document which is hopefully more convenient for the reviewers.
>
> While the submitted main paper linked explicitly to them *“we provide further details into the propagation of light through optical fibers and how we construct theoretical TMs in Appendix A.4 and further details in the inversion of the TM in Appendix A.5”*. We had this information in the appendix as we assumed it was well accepted and dates back to the seventies, see Gloge, 1971. Although, given the extra page allowed before the final paper we will transfer some of this explanatory background to the main body of the paper to make the rationale clearer to readers.
>
> If these two sections in the appendix do not address your concerns please let us know and we will be happy to provide further clarification.
>
> D. Gloge, "Weakly Guiding Fibers," Appl. Opt.  10, 2252-2258 (1971)

---

> > ### Author Response · Authors · 2022-08-01
> > **Response to Reviewer (2/3)**
> >
> > * The only experimental validation is on (F)MNIST which isn't convincing.
> >
> > This very brief comment does not make clear what is not convincing about these results. We interpret the comment on FMNIST as not being convincing as being related to the perceived simplicity of FMNIST as a task (but if we have misunderstood this, please detail what is not convincing (what does the reviewer think is erroneous in the results), and what would be needed to convince them (bearing in mind the state of the art in this field)?
> >
> > Our argument that FMNIST is not overly simple to use, is that in this paper we are not trying to classify these images (which is clearly a simple task), we are using them as a test of our reduced parameterisation optical fiber calibration process. If we were suggesting a more complex model (e.g. a multi-layer VAE) then the low-resolution images could well be argued to be vulnerable to overtraining if the model were over-parameterised. However, we are able to demonstrate that the vastly reduced parameterisation of the Bessel representation, in line with the physics of the fibre, works both on well-controlled, simulated, but much higher resolution images than the current state-of the-art, as well as on the lower-resolution lab experiments common in papers in the field as the current state of the art (published at NeurIPS by Moran et al. 2018, and in Nature Communications by Caramazza et al 2019), and which are therefore a sensible baseline for this work.
> >
> > As further evidence of the rigor of our experimental work, in the appendix we report a series of tests of the robustness of the  framework to changes in the quality of the available data:
> > A.6 Results for Accounting for Losses in a Real System - testing diagonal assumption
> > A.7 Effects of Noise on Speckled Images - we compare the effect of noise on the speckled images when using a theoretical TM.
> > A.14 Impacts of Reducing Training Dataset Size
> > At the request of another reviewer we have added: A.15 Impacts of Under Parameterising the set of Bessel Function Bases - which investigates sensitivity of the result to using the full Bessel basis.
> >
> > In summary, we tested the method on two different fibre models, we show with the loss values that our method performs well and visually that our method clearly produces the correct objects in both perfectly controlled simulations, and with real world experimental data, and in the appendix we systematically test the process with a range of conditions, and we provide executable code and data for others to use, so we believe that this contribution has had a suitable experimental validation to be a meaningful contribution at this stage, for others to build on.
> >
> >
> >
> > * The simulated speckle data (Fig 6) looks considerably more structured than the experimental data (Fig 5), suggesting the adopted model is inaccurate.
> >
> > To provide some clarification, the results in Fig 3 in  section “4.1 Real Multimode Fibre” are the results using the experimental lab data (Moran et al. 2018) are produced using a fiber with ~8000 modes taking as input 28x28 images and producing 224x224 speckled patterns, and the results in the section “4.2 Theoretical TM" are the results using simulated data based on a completely different fiber. In figs 4 and 5 in section “4.2 Theoretical TM" we use simulated data produced with a fibre with ~1000 modes taking as input 28x28 images and producing 180x180 speckled patterns, and the simulated data for fig 6  in section “4.2 Theoretical TM" is produced using a fiber with ~1000 modes taking as input 256x256 images and producing 256x256 speckled pattern. Fig 3 is experimental data and Figs 4, 5, and 6 are simulated data. (See Appendix A.4 in the paper for details on the simulated TMs, and (Moran et al. 2018) for details on their experimental set-up.)
> >
> > Considering the use of different fibres, it is expected that the speckled patterns could look different. However, you can see other papers in the published literature showing speckles in experiments  which show both types of speckle structure visible in our paper, from the same MMF, depending on illumination, e.g. Figure 3 of https://opg.optica.org/oe/fulltext.cfm?uri=oe-20-10-10583&id=232812 which suggests that the speckles visible in our work appear very similar to those obtained in other labs for other fibres and experimental conditions.
> >
> > * Some details are missing. It seems the paper is considering only "incoherent" fiber bundles.
> >
> > There is a misunderstanding here as we are **not** using fiber bundles. We are considering a single multi-mode fiber, as stated in 2.1 ‘Multi-mode fibres present a clear advantage over single-mode fibre bundles due to having 1-2 orders of magnitude greater density of modes than a fibre bundle’.

---

> > > ### Author Response · Authors · 2022-08-01
> > > **Response to Reviewer (3/3)**
> > >
> > > * There's no evidence the proposed method generalizes across MMFs.
> > > * Finally, to address the limitation concerns: Not discussed. Method is specific to a particular MMF. Moreover, if that fiber bends (like it would in endoscopy) that fibers TM is likely to change.
> > >
> > >
> > > The task of generalizing across MMFs is not the task we attempted to solve within the paper. As each MMF has a different transmission matrix, there is no simple solution to this, and we make no claims to have solved that problem. We agree that bending in fibers is an unsolved problem, although it is not the problem attempted in this paper. Instead we developed a physics-informed model that reduces the parameter count required to be learned which we show can scale to higher-resolution images than previous methods and due to the massive reduction in parameters  can learn with fewer training data points than previous methods.
> > >
> > > In the case of generalizing to new TMs  which are the same fiber but in a different configuration, such as after bending, we believe that our proposed architecture will have benefits in addressing this due to fiber bending having previously been shown to amount to mode mixing within the TM (Resisi et al, 2020), so TMs tend to change in a somewhat continuous manner.
> > >
> > > This means that modifying our diagonal complex weight matrix (rather than the entire transmission matrix) according to inferred bend context is a potentially powerful future method to address TM-change due to bending, as it gets to the core of how the TM  changes. The reduction of parameters and faster learning can be used in networks that bring multiple configurations together, as we would then be interpolating between matrices scaling as O(N_modes) rather than O(N^4) for NxN images. Hence, one of the main intended benefits of our proposed architecture is in fact  that it will help to  address  the challenge of changing TMs in bending fibres.
> > >
> > > We would be happy to make this clearer in the final version and  add this in as a limitation, with implications in a  future work section, as we intend to build on this paper’s results to address bending  in future papers, allowing progress to future systems which could generalize to new MMF configurations. (But a completely new MMF will always require calibration)
> > >
> > > Resisi, S., Viernik, Y., Popoff, S.M. and Bromberg, Y., 2020. Wavefront shaping in multimode fibers by transmission matrix engineering. APL Photonics, 5(3), p.036103.
> > >
> > > * There are minor inaccuracies in the paper: Line 270 states reducing the calibration dataset size minimizes how long cells have to be imaged; cells are involved in the calibration of a TM.
> > >
> > > Thank you for highlighting this line. While we are not quite sure what you mean by the cells being involved in the calibration (BTW - there is a typo on L270 though as it should read for imaging cells rather than as imaging cells), our intention had been to highlight examples where reducing the amount of training data could be beneficial in more than just time (requiring lots of physical samples to image because the repeated imaging required of a large dataset would damage physical samples is ameliorated by using our method).
> > >
> > > However, on re-reading we can see that it is clear that a lot more explanation of the specifics of the details of the end use case would be needed to avoid this being confusing, so it might be better for us to cut out this line to improve the focus and clarity of the paper, as none of the main claims or applications requires us to discuss this point in this paper.
> > >
> > > * Furthermore, to address the question: How would fitting a low rank approximation of the TM (or its inverse) compare to the proposed method?
> > >
> > > Fitting a low rank approximation of the TM would require the TM to be known, which is not the case for a new fiber. There are works which seek to characterize the TM for a fiber, which are cited in Section1 “Introduction”, but these often require expensive experimentation and a large amount of data collection. Once this was known, a low rank approximation of the TM could be fit, which would reduce the computational expense of holding a TM in memory and performing the large matrix multiplication required to pass an image through a TM.
> > >
> > > Our method is arguably an example of a low rank approximation of the matrix, informed by the physics of the problem. You may be interested in the results of the new section A.15 added to the appendix which shows how performance changes as you remove further bases from the model.

---

> > ### Comment · Reviewer_iftp · 2022-08-07
> > **Concerns mostly addressed**
> >
> > Thank you for your very detailed response and for pointing me to A4 and A8; these sections have addressed the most serious of my concerns.
> >
> > There was a typo in my review: "cells are involved in the calibration of a TM" should have read "cells are not involved in the calibration of a TM." To my knowledge, TM estimation is generally performedas a calibration/training step before imaging and is not done in vivo. Thus the claim that the proposed method would reduce cell bleaching by accelerating the training process seems unlikely.
> >
> > My concern about (F)MNIST being toy datasets is that these are very low dimensional datasets (far less than R^{28x28}). Accordingly NN backend may easily learn to reconstruct these images even if Bessel functions are a poor representation for the TM

---

> > > ### Author Response · Authors · 2022-08-08
> > > **Response to Reviewer**
> > >
> > > Thank you for your response! We are happy those sections addressed your most serious concerns.
> > >
> > > Also, thank you for raising the point on cells. We were hoping to give an application where requiring a smaller training dataset would be a benefit, if a future system were to be re-calibrated in situ (with a specific bend). On reflection, we have decided to remove this point, as further details (about, e.g. a significantly more complicated VAE architecture using our BEM method at its core) would be required to avoid the confusion, complicating the message of the paper. As none of the main claims made in the paper depend on this point, we now think it will be cleaner to remove it. Thank you for helping us refine our core message!
> > >
> > > We now better understand your concern around the fMNIST dataset. Your concern is very valid for previous complex linear and CNN based approaches, and we believe this has been an issue in some of the previous published work. However, our approach of using an efficient physics motivated model is designed to avoid overtraining, by having significantly fewer trainable parameters in the core BEP model. We then added the post processing model to tidy and sharpen the images from the output of this model. The final sentence of Sec 3.2 explicitly states this danger of bias to the prior, and the need to design interfaces for safety-critical applications around this issue.
> > >
> > > In our table of results and figures (Fig 3, 4, 5 and 6) we always included both the output of the physics motivated model (BEM)  and then that model combined with the post processing model (BEM+PP)  so that the contribution of both of these can be seen. We also show that our model trained on fMNIST generalises well to MNIST in Figure 5, which highlights that our physics motivated model has not over-fit only to fMNIST and we provide the results on the higher resolution Imagenet images in section 4.2.2. Finally, the new section we added A.15 shows how performance changes as some bases are removed from the model. We acknowledge your point though about the dangers of using particularly low resolution images, although we felt the need to use these to compare to previous published SOTA methods, and we can add in the paper that while our high-resolution simulated results are very promising, future work should utilise higher resolution images in lab experiments for further validation.

---

### Official Review · Reviewer_cTsz · 2022-07-12

**Rating:** 6
**Confidence:** 4
**Soundness:** 3 good
**Presentation:** 3 good
**Contribution:** 3 good

**Summary:**

This paper proposes to use a bessel basis and a post-processing neural network for modeling the optical transmission through a multi-mode optical fiber. The key idea is that multi-mode fiber outputs are band-limited with respect to the bessel functions of different modes. Applying this representation for the multi-mode fiber output allows us to reduce the number of parameters significantly, reducing the computational burden of inverse estimation: from a speckle image to a target natural image.


**Questions:**

My concern mainly lies in the analysis of the representation capability of the bessel basis. How accurate the basis is for modeling the MM fiber outputs, especially w.r.t. the number of bases used? What are the remaining components that cannot be modeled? Probably visualizing the full weight matrix might be useful to answer these questions.

Second, many works using MM fibers for display applications modulate incident wavefront to generate high-fidelity images directly out of the fiber. Could the proposed method be also used for this application?

**Limitations:**

The authors adequately described the limitations.

**Strengths And Weaknesses:**

Strengths
- For the first time, a 256x256 image is reconstructed from a speckle image, thanks to the efficient parameterization.
- Using the bessel representation is a neat idea for the neural modeling of the multi-mode fiber outputs.
- The paper is well written.

Weaknesses
- The authors show that the bessel representation is effective in modeling the MM fiber output. However, there is no analysis on it's representation capability with respect to the number of bases.

---

> ### Author Response · Authors · 2022-08-01
> **Response to Reviewer**
>
> Thank you for your review and acknowledging that our new model allows 256x256 image reconstruction for the first time, that the idea is interesting, and that our paper is well written.
>
> We will address the concerns below:
>
> * The authors show that the bessel representation is effective in modeling the MM fiber output. However, there is no analysis on it's representation capability with respect to the number of bases.
> * My concern mainly lies in the analysis of the representation capability of the bessel basis. How accurate the basis is for modeling the MM fiber outputs, especially w.r.t. the number of bases used? What are the remaining components that cannot be modeled? Probably visualizing the full weight matrix might be useful to answer these questions.
>
> In the paper (in 4.1) we discuss some mismatches between the basic theory of Bessel functions and the  reality for a specific fiber (e.g. sharp bends, dopant diffusion, elliptical cores), and the paper explored non diagonal matrices (Sec 4.1) and post-processing layers (sec 3.2) for this reason.
> Visualizing a full Transmission Matrix is not usually easy to interpret directly, e.g. for 256 x 256 images we would have a 65536 x 65536 matrix (4,294,967,296 elements).
> Despite this, the reviewer makes an interesting suggestion regarding sensitivity to the number of bases used. We can explore the representation capability with respect to the number of bases, although it is also worth noting that there is usually a natural choice for the number of bases and this is driven by the fiber being used. We have performed the extra analysis requested, and added a new section in the appendix (the new A.15), which you can see in the attached updated pdf. This shows the impact of reducing the number of bases used for the task of reconstructing fMNIST images. Here we consider progressively removing high frequency bases by removing the highest frequency radial bases. We show results with the full basis set which comprises 21 radial frequencies (1061 bases) and reduced basis sets which comprise 14 radial frequencies (932 bases), 7 radial frequencies (567 bases), and 4 radial frequencies (322 bases). We hope the addition of this new section alleviates your concerns by showing that if the basis set is under-parameterised, such that high frequency bases are missing, the method can still achieve good reconstructions, although some higher frequency information is lost.
>
>
> * Second, many works using MM fibers for display applications modulate incident wavefront to generate high-fidelity images directly out of the fiber. Could the proposed method be also used for this application?
>
> In principle this is possible. In an idealized fiber, as is the case with the simulated transmission matrix, it would be a simple case of adding output spots as needed for the image. For real data, the phase between spots will play a role by producing an interference between neighboring output spots. This can be iteratively mitigated with knowledge of the full transmission matrix as shown in Figure 6 in work by Cizmar and Dholakia https://opg.optica.org/oe/fulltext.cfm?uri=oe-19-20-18871&id=222508. However, as far as this work has investigated, it remains unclear as to whether enough information about the transmission matrix is collected to do this projection. We appreciate the interesting suggestion.

---

### Meta-Review · Area_Chair_5Gwy · 2022-09-01

**Recommendation:** Accept
**Confidence:** Certain

**Metareview:**

This paper proposes a new learning-based technique for imaging through multimodal fibers (MMF). A key idea of this work is to exploit the property that the transmission matrices associated with MMFs are approximately diagonalizable by Bessel basis. This idea then allows one to significantly reduce the number of parameters, which in turn reduces the memory usage and computational burden of the inverse problem of recovering a natural image from speckle patterns. The authors demonstrated the effectiveness of their proposed algorithm on low-resolution experimentally captured data as well as on higher-resolution simulated dataset.


**Award:**

No

---

### Decision · Program_Chairs · 2022-09-14

Accept